# A new view of transcriptome complexity and regulation through the lens of local splicing variations

Jorge Vaquero-Garcia[1,2†], Alejandro Barrera[1,2†], Matthew R Gazzara[1,3†], Juan Gonzalez-Vallinas[1,2], Nicholas F Lahens[4], John B Hogenesch[4], Kristen W Lynch[1,3], Yoseph Barash[1,2*]

[1]Department of Genetics, Perelman School of Medicine, University of Pennsylvania, Philadelphia, United States; [2]Department of Computer and Information Science, University of Pennsylvania, Philadelphia, United States; [3]Department of Biochemistry and Biophysics, Perelman School of Medicine, University of Pennsylvania, Philadelphia, United States; [4]Department of Pharmacology, Perelman School of Medicine, University of Pennsylvania, Philadelphia, United States

**Abstract** Alternative splicing (AS) can critically affect gene function and disease, yet mapping splicing variations remains a challenge. Here, we propose a new approach to define and quantify mRNA splicing in units of local splicing variations (LSVs). LSVs capture previously defined types of alternative splicing as well as more complex transcript variations. Building the first genome wide map of LSVs from twelve mouse tissues, we find complex LSVs constitute over 30% of tissue dependent transcript variations and affect specific protein families. We show the prevalence of complex LSVs is conserved in humans and identify hundreds of LSVs that are specific to brain subregions or altered in Alzheimer's patients. Amongst those are novel isoforms in the Camk2 family and a novel poison exon in Ptbp1, a key splice factor in neurogenesis. We anticipate the approach presented here will advance the ability to relate tissue-specific splice variation to genetic variation, phenotype, and disease.

*For correspondence: yosephb@ mail.med.upenn.edu

†These authors contributed equally to this work

**Competing interests:** The authors declare that no competing interests exist.

## Introduction

Production of distinct mRNA isoforms from the same locus has been shown to be common phenomena across metazoans (*Barbosa-Morais et al., 2012*; *Merkin et al., 2012*). Different isoforms may arise through the use of alternative transcription start and end sites, or through alternative processing of pre-mRNA. A key process is alternative splicing (AS) of pre-mRNA, where different subsets of pre-mRNA segments are removed while others are joined, or spliced together. The resulting differences between the mature mRNA isoforms can, in turn, encode different protein products, or affect mRNA stability, localization, and translation. Over 95% of human multiexon genes undergo AS, and disease associated genetic variants have been shown to frequently lead to splicing defects (*Cooper et al., 2009*; *Pan et al., 2008*; *Wang et al., 2008*). These observations emphasize the need to accurately map and quantify splice variations.

RNA-Seq technology has advanced the detection and quantitation of splice variants by producing millions of short sequence reads derived from the transcriptome. Despite constant technological advancement, the combination of limited coverage depth, experimental biases, and reads spanning only a small fraction of the variable parts of transcripts has left accurate mapping of transcriptome variations an open challenge (*Alamancos et al., 2014*).

**eLife digest** Genes contain coded instructions to build other molecules that are collectively referred to as gene products. Building these products requires the gene's instructions to be copied into a molecule of RNA in a process called transcription. Over 90% of human genes undergo a process by which different segments of the transcribed RNA molecule are either removed or retained. This process, termed alternative splicing, results in a single gene encoding different gene products that can perform in different ways.

Alternative splicing can also mean that gene products vary between different cells, tissues and individuals. Some of these variations can be harmful and lead to disease. However, it is difficult with current methods to accurately identify variations in gene products that are due to alternative splicing and see how these products differ between groups of people, such as patients and healthy controls.

Vaquero-Garcia, Barrera, Gazzara et al. have now developed new methods to define, measure and visualize the variations in RNA gene products. First, splicing variations were catalogued across a range of species from lizards to humans, which revealed that some fairly complicated variations were much more common than previously appreciated. These complex variations had not been studied much before, but the new methods showed that they make up a third of the variations in the RNA products copied from human genes.

Vaquero-Garcia, Barrera, Gazzara et al. then showed that the new methods are more accurate and sensitive than previous methods, and can be used to discover splicing variations that were previously unknown. For example, applying the new methods to data collected in other studies revealed variations in genes that are important for brain development and activity. Further analysis then showed that these variations were also altered in brain samples from patients with Alzheimer disease.

The new methods developed by Vaquero-Garcia, Barrera, Gazzara et al. can now shed new light on gene product variations, especially the more complex ones that have not been studied before. The next challenge is to use these tools to better understand the regulation and purpose of splicing variants and how they can contribute to diseases in humans.

Transcriptome variations have been traditionally studied either at the level of full gene isoforms or through the specification of alternative splicing 'events'. The latter have been categorized into several common types, such as intron retention, alternative 3'/5' splice sites, or cassette exons. Importantly, while exact isoforms and their quantifications cannot be directly inferred from the short RNA-Seq reads, AS events can be detected via reads that span across spliced exons (junction reads). Both AS events and full isoforms can be captured by a gene schematic or a splice graph (*Heber et al., 2002*), where edges (lines) connect pre-mRNA segments spliced together in different transcripts (*Figure 1A*, top).

While useful, the previously defined AS types fail to capture the full complexity of spliceosome decisions. Specifically, AS types represent spliceosome decisions as strictly binary, involving only two exons or two splice sites in the same exon. The bottom panel in *Figure 1A* illustrates a few possible splicing variations that do not fit the previously defined AS types and can involve more than two alternative junctions.

*Figure 1B* serves as a visual summary for both the potential and challenges in analyzing splicing variations. Combining known transcripts and RNA-Seq data results in the *Camk2g* splice graph shown (*Figure 1B*, top). This splice graph includes novel, unannotated, splice junctions detected from junction spanning RNA-Seq reads (green), as well as a complex case where exon 14 can be spliced to exons 15, 16, or 17 (*Figure 1B*, middle). Quantification by RT-PCR in several mouse tissues validate the existence of these variations and also points to isoforms that are predominantly produced in brain subregions and in muscle (*Figure 1B*, bottom). In order to achieve such results we need to have a computational framework that efficiently combines RNA-Seq with existing gene annotation and enables us to accurately detect, quantify, and visualize diverse splicing variations across different experimental conditions.

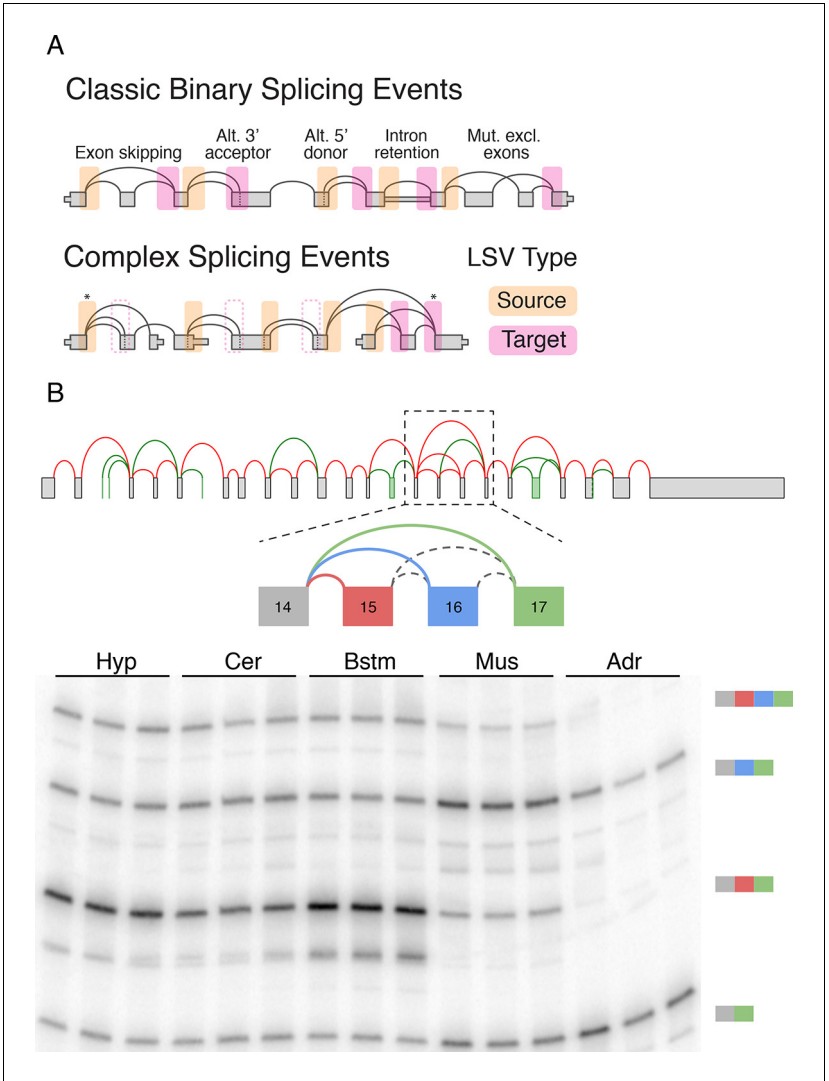

**Figure 1.** LSV formulation and prevalence. (**A**) LSVs can be represented as splice graph splits from a single source exon (yellow) or into a single target exon (pink). LSV formulation captures previously defined, 'classical', binary alternative splicing cases (top) as well as other variations (bottom). An asterisk denotes complex variations involving more than two alternative junctions; dash line denotes redundant LSVs that are a subset of other LSVs (see Materials and methods). (**B**) Example of a complex LSV in the *Camk2g* gene. The gene's splice graph (top) includes known splice junctions from annotated transcripts (red) and novel junctions (green) detected from RNA-Seq data. The splice graph includes a complex LSV involving exons 14–17 (middle). RT-PCR validation of the LSV in brainstem, cerebellum, hypothalamus, muscle, and adrenal is shown at the bottom. Several isoforms are preferentially included in brain and muscle.

## Results

### Formulation of local splicing variations (LSVs)

To address the shortcomings of previously defined AS types we suggest the formulation of local splicing variations, or LSVs. LSVs are defined and easily visualized as splits (multiple edges) in a splice graph where several edges either come into or from a single exon, termed the reference exon. A Single Source (SS) LSV (*Figure 1*, yellow) corresponds to a reference exon spliced to several downstream RNA segments while single target (ST) LSV (*Figure 1*, pink) corresponds to a reference exon spliced to upstream segments. The full specification of an LSV also includes the relative location of the exons and junctions (see Material and methods). *Figure 1A* illustrates how this formulation

captures previously defined AS types (top panel) as well as more complex cases (bottom panel). Specifically, previously defined 'classical' AS events appear as special cases of binary graph splits (e.g., include or skip a cassette exon), while LSVs capture non-classical binary splits and splits involving more than two junctions. Such non-binary splits are termed complex LSVs. LSVs can also involve intron retention (intronic LSVs) or be comprised of only exons (exonic LSVs). Moreover, the transcriptome variability captured by LSVs may be the result of not only spliceosome decisions but also of alternative transcription start or end positions. For example, the gene in *Figure 1A* bottom panel involves two alternative first exons so a relative change in the transcription start site usage can result in changes in downstream LSVs quantification. Importantly, LSV formulation allows the probing of transcriptome structure and complexity yet, unlike full transcripts, can still be quantified directly from junction spanning reads.

## LSV detection, quantification and visualization using MAJIQ

In order to address the challenges involved in detection, quantification and visualization of LSVs we developed a new computational framework that we have termed Modeling Alternative Junction Inclusion Quantification (MAJIQ). MAJIQ's first step (*Figure 2A*, top) is to parse a known database of transcripts, given as a GFF3 annotation file, along with a set of mapped and aligned RNA-Seq experiments (indexed BAM files). Unlike many methods that only analyze known isoforms, MAJIQ supplements known transcripts with 'reliable' edges derived from *de novo* junction spanning reads. Several filters can be applied to define which edges are considered reliable and which LSVs have enough reads to be later quantified (see Material and methods). Similarly, LSVs whose edges are a subset of other LSVs, such as those denoted with dashed rectangles in *Figure 1A*, are removed to avoid redundancy (see Material and methods). Next, MAJIQ can be executed to quantify LSVs either in a specific condition or to compare two experimental conditions, with or without replicates. LSV quantification in a specific condition is based on the marginal percent selected index (PSI, denoted Ψ) for each junction involved in the LSV, while comparison of experimental conditions is based on relative changes in PSI (dPSI, ΔΨ). MAJIQ uses a combination of read rate modeling, Bayesian Ψ modeling, and bootstrapping to report posterior Ψ and ΔΨ distributions for each quantified LSV. The results of MAJIQ's LSV detection and quantification can be interactively visualized with the package VOILA in a standard web browser (*Figure 2A* bottom).

We assessed MAJIQ's quantification accuracy for both Ψ and ΔΨ using a combination of RNA-Seq from biological replicates and an extensive set of 208 RT-PCR validations. These experiments included two mouse tissues (cerebellum and liver [*Zhang et al., 2014*]), and a human Jurkat T cell line (unstimulated and stimulated, [*Cole et al., 2015*]). While accuracy depended on the dataset used, MAJIQ achieved an overall correlation of R = 0.8 and R = 0.95 for PSI and dPSI quantification by RT-PCR, comparing favorably to alternative methods on all datasets (*Figure 2B,C*, *Figure 2—figure supplement 1*). Next, we used biological replicates from the Mouse Genome Project (*Keane et al., 2011*) to assess reproducibility of differential splicing detection from RNA-Seq when comparing two experimental conditions. The reproducibility ratio (RR, see Material and methods) captures the fraction of top ranked differentially spliced LSVs that maintain their top ranking when analyzing another set of replicate experiments. *Figure 2D* shows MAJIQ compares favorably to other methods, including MISO (*Katz et al., 2010*), rMATS (*Shen et al., 2014*), and a bootstrapping approach (*Xiong et al., 2015*) adopted for LSV. While MISO and rMATS achieved a reproducibility ratio of 61–67% we found the bootstrapping approach (N.B.) suffered from particularly high variance, which degraded reproducibility of LSVs ranking. In comparison, MAJIQ achieved a mean RR=77% when comparing two pairs of experiments and improving to RR=86% when the experiments compared had replicates. Notably, detection power was also improved. Defining differentially spliced LSVs as those for which $P(|\Delta\Psi|>0.2) > 0.95$, the number of detected LSVs (N), after removing LSVs overlap (see Materials and methods), was on average 400 for pairwise and 447 for group comparisons, compared to 240 and 260 respectively by rMATS. The improvement in both detection and reproducibility of differentially spliced LSVs (N, RR) was robust to the statistical threshold used to define *N* (*Figure 2—figure supplement 2A*) and when we removed MAJIQ's de-novo junction detection the number of LSVs dropped as expected but reproducibility remained high (N = 337, RR= 87%, data not shown). Importantly, this result also indicated that including de-novo junctions increased the number of differentially spliced LSVs that could be detected by over 30% (337 vs. 447), while retaining equivalent reproducibility. Defining differential splicing reproducibility by RT-

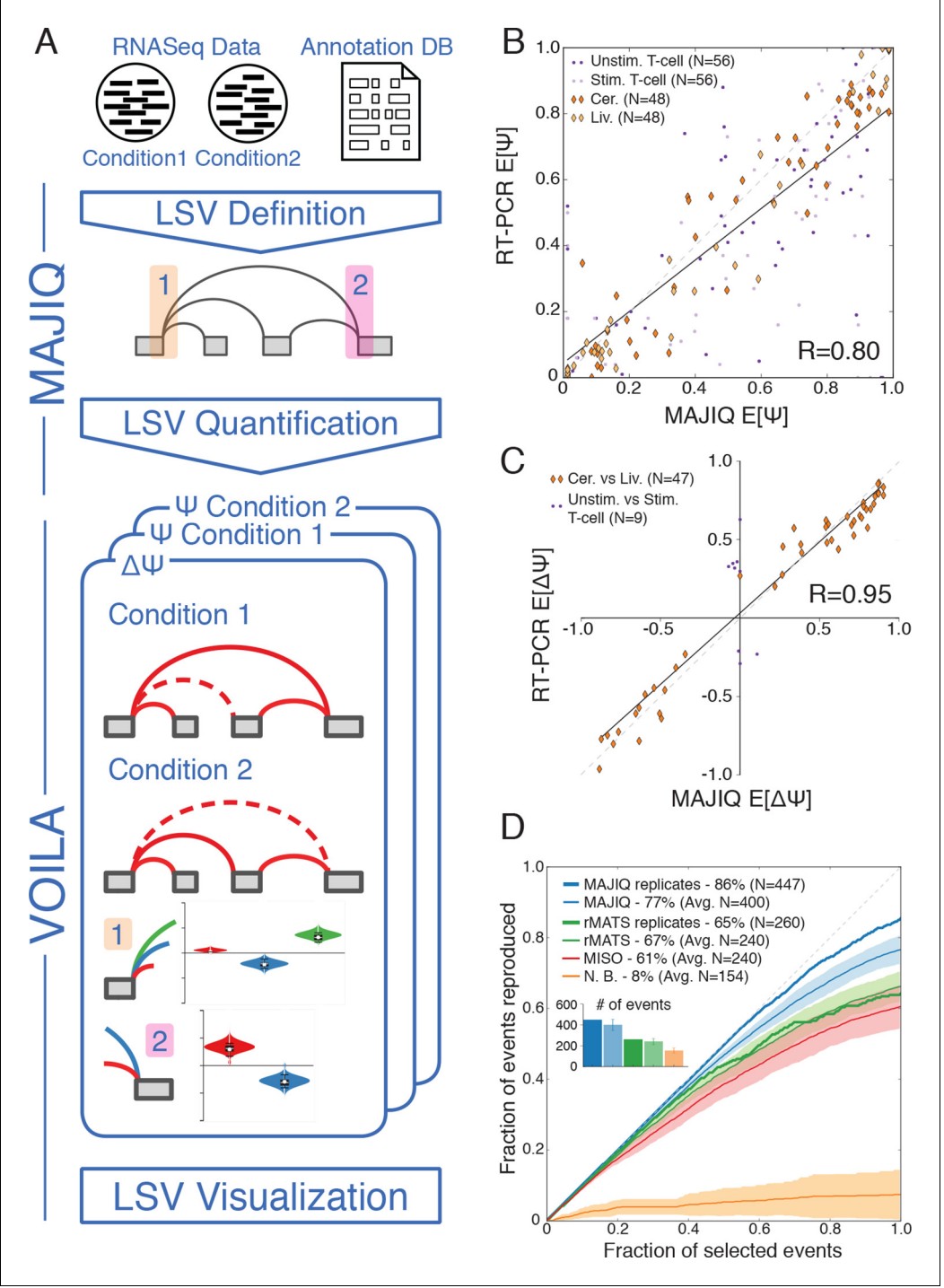

**Figure 2.** LSV analysis using MAJIQ. (**A**) MAJIQ's analysis pipeline. RNA-Seq reads are combined with an annotated transcriptome to create splice graphs and detect LSVs for each gene, then LSVs are quantified and compared between conditions. The visual output (VOILA) lists LSVs with violin plots representing estimates of percent inclusion index (PSI, $\Psi$) or changes in inclusion (dPSI, $\Delta\Psi$). Two cases are illustrated, for a single source three way LSV (orange), and a single target two way LSV (pink). (**B**) Correspondence between E[$\Psi$] by MAJIQ and $\Psi$ by RT-PCR. R is the correlation coefficient. Colors and shapes represent different experimental conditions: mouse cerebellum and liver (dark and light orange diamonds, respectively); human unstimulated and stimulated T-Cells (dark and light purple dots, respectively). Total n = 208. (**C**) Correspondence between E[$\Delta\Psi$] by MAJIQ and $\Delta\Psi$ by RT-PCR, where $|\Delta\Psi^{RT}|>0.2$. R is the correlation coefficient. Changes in inclusion were measured between liver and cerebellum mouse tissues (diamonds, n = 45); stimulated and unstimulated T-Cells (dots, n = 9). (**D**)
*Figure 2 continued on next page*

*Figure 2 continued*

Reproducibility ratio (RR) of high confidence differentially included LSVs, *i.e.* LSVs for which P(|ΔΨ|> 0.2) > 0.95), when comparing RNA-Seq from two conditions. A differentially included LSV is considered replicated if it maintains a rank at least as high as N in biological replicates, where N is the set size. LSVs are ranked by E[ΔΨ] and filtered for overlap. Twelve replicate pairs from *Keane et al. (2011)* were used to compute the histogram's std (light blue). Other lines show MAJIQ's RR with replicates (thick blue), RR for AS events detected by rMATS w/ wo replicates (light and dark green), MISO (red), and RR for LSVs using Naïve Bootstrapping (orange). The inset bar chart shows the number of LSVs or AS events (N) derived by each method and used in the RR plots (see Materials and methods for more details).

The following figure supplements are available for figure 2:

**Figure supplement 1.** Quantifying PSI and dPSI accuracy.

**Figure supplement 2.** Quantifying differential splicing reproducibility.

---

PCR as LSVs for which $|\Delta\Psi^{RT}|$>20% resulted in 95% reproducibility. The higher reproducibility by RT-PCR can be expected given the lower experimental variability compared to RNA-Seq. Notably, the LSVs tested by RT-PCR were selected to cover a wide spectrum of read depth. We found that while higher coverage allowed more differential LSVs to be detected and steadily increased reproducibility by RNA-Seq, MAJIQ's reproducibility by RT-PCR was stable across read coverage depth, pointing to the robustness of the method (*Figure 2—figure supplement 2B*). Finally, we note that the above RT-PCR evaluation concentrated on binary LSVs to allow comparison to currently available methods, but we observed similar accuracy for the quantification of complex LSVs (*Figure 2—figure supplement 2C*).

## Complex LSV are prevalent in diverse metazoa

To assess the significance of LSVs formulation we estimated LSVs prevalence in several metazoans, ranging from lizard to human (*Figure 3*). Naturally, this analysis is affected by how well a species transcriptome is annotated, and how permissive the database used is. In human for example, complex LSVs constitute 20.6% to 33.7% of the LSVs in annotated transcripts by RefSeq and Ensembl respectively, but only 1.86% in opossum's Ensembl annotation (*Figure 3A,B*). Next, we expanded the set of annotated transcripts with novel junctions detected from RNA-Seq junction spanning reads. Limiting our analysis to only 5–6 similar tissues in all species and conservative junction detection still increased the total number of LSVs in human by 11% and the fraction of complex LSVs from 33.7% to 37.1% (*Figure 3A*). In species not as well annotated the effect of adding RNA-Seq data was more dramatic, jumping in opossum for example from 1,610 to 10,228 LSVs, of which 10% were complex. In summary, while LSV analysis across species was confounded by read coverage and transcriptome annotation we find that non-classical and complex LSVs make up a substantial fraction of observed transcriptome variations. Such complex LSVs are likely to be removed, undetected, or mislabeled by algorithms that only quantify binary AS events from previously annotated transcripts.

## A genome wide view of LSV across 12 mouse tissues

Given the clear impact of the RNA-Seq dataset and transcriptome annotation, we chose to focus our genome wide analysis on a recent mouse dataset. This allowed us to analyze 12 tissues with an average of over 120M reads per sample, produced by a single lab (*Zhang et al., 2014*). This data included three brain subregions, eight samples per tissue, and matching RNA for RT-PCR validations, leading to a total of 100,512 LSVs detected. First, we used this data to assess the usage of LSVs across tissues. In order to minimize LSVs that result from false junctions identified by the mapper we only included junctions with multiple uniquely mapped staggered reads across multiple biological replicates (see Material and methods). Next, we tested the maximal inclusion level of the second, third, or the least used junction in an LSV across twelve mouse tissues. We detected a switch behavior where a different junction becomes dominant at 50% inclusion or more in approximately 5% of the classical binary LSVs (*Figure 4A*, grey), compared to 12% for the second most used junction in complex LSVs (*Figure 4A*, light green). Setting a conservative threshold of Ψ > 10% to

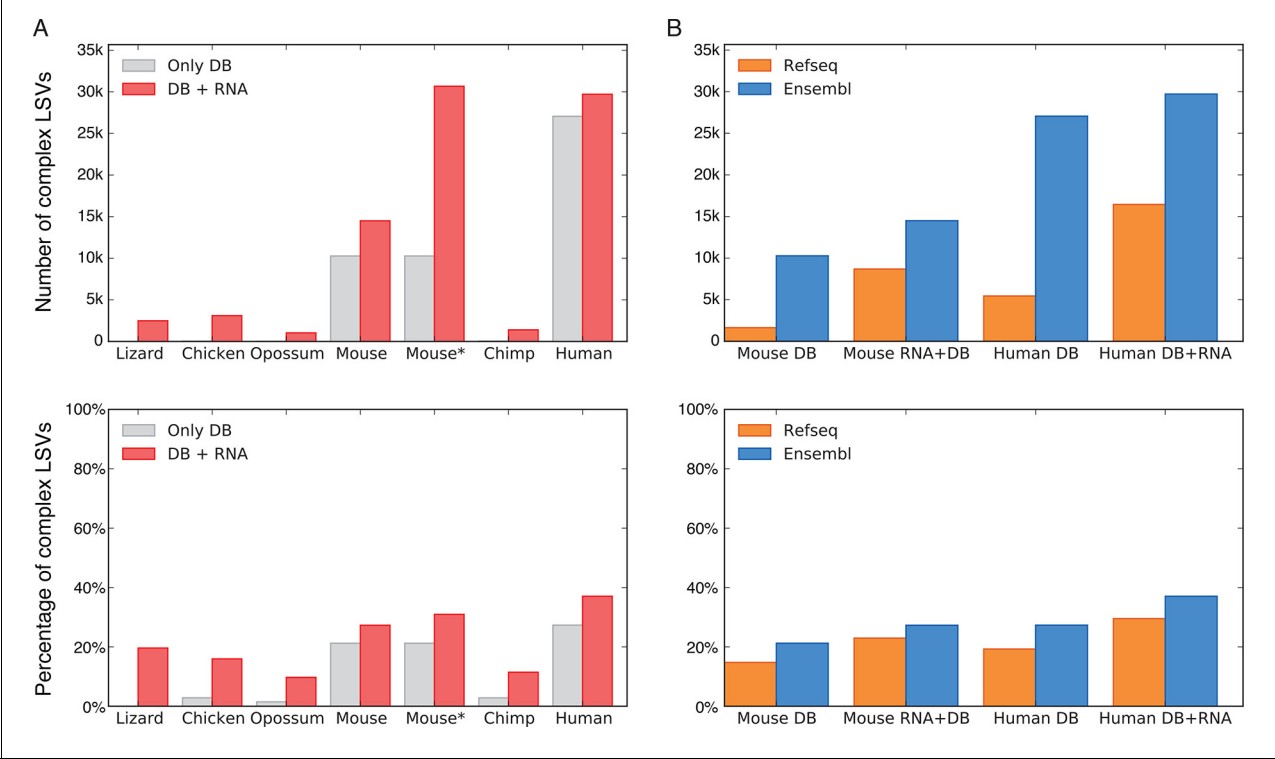

**Figure 3.** LSV prevalence across diverse metazoans. (**A**) Number of LSVs (top) and fraction of complex LSVs (bottom) when using Ensembl annotated transcripts only (grey) or combining it with RNA-Seq from 5–6 similar tissues (red). Mouse* is the dataset from *Zhang et al. (2014)*. (**B**) Number of LSVs (top) and fraction of complex LSVs (bottom) when using RefSeq (orange) and Ensembl (blue). The RNA-Seq dataset is the same as in (**A**).

denote splice junctions that are less likely to be splicing noise or database errors we find that for the classical binary LSVs approximately 32%, or 9,516 pass that threshold, compared to 57% and 19% of the complex LSVs that pass that threshold for the second and third most used junction respectively. These correspond to a total of 6,338 and 2,112 LSVs in our datasets, pointing to the importance of complex LSVs in transcriptome analysis. Even when testing for the least used junction in complex LSVs (e.g. the ninth in a nine junction LSV), we still find almost 10% pass the 10% inclusion threshold (*Figure 4A*, dark green). Finally, for intronic LSVs we find almost 11,000 cases where an intron is retained at least 50% in one tissue, and 3,844 cases where the intron is almost always retained with $\Psi > 99\%$ (*Figure 4—figure supplement 1D*). This observation of widespread intron retention (IR), especially in brain tissues, is in line with a recent study across many more tissues and cell lines (*Braunschweig et al., 2014*), though our overall estimate of IR prevalence is more conservative.

Commonly occurring network substructures, or network motifs, have garnered much research attention in diverse fields (*Milo et al., 2002*). Gene splice graphs can also be thought of as networks with exons as nodes and spliced junctions as edges. In this interpretation, LSVs can be thought of as small network motifs and used to shed light on the transcriptome complexity and commonly reoccurring sub-structure. Comparing the frequency of exonic LSV types (*Figure 4B*) we find that the more common non classical LSVs involve 3 to 5 exons, combine exon skipping with an alternative 3'/5' splice site, or involve alternative transcript start/end at the LSV's reference exon. In contrast, intronic LSVs are much less diverse, with classical intron retention making 68% of the cases (*Figure 4—figure supplement 1C*). *Figure 4C* shows that for exonic LSVs 14% involve more than 2 exons, 30% of the single source and 20% and of the single target LSVs involve a reference exon with two or more 5'/3' splice sites, respectively. Overall, complex (non-binary) LSVs comprise 36.2% of the transcriptome variations detected in the data and 27.5% of the variations deemed quantifiable (see Materials and methods), yet spliceosome decisions still appear localized, with few LSVs involving more than 6 exons or junctions. When analyzing LSVs usage, we found that the biochemical

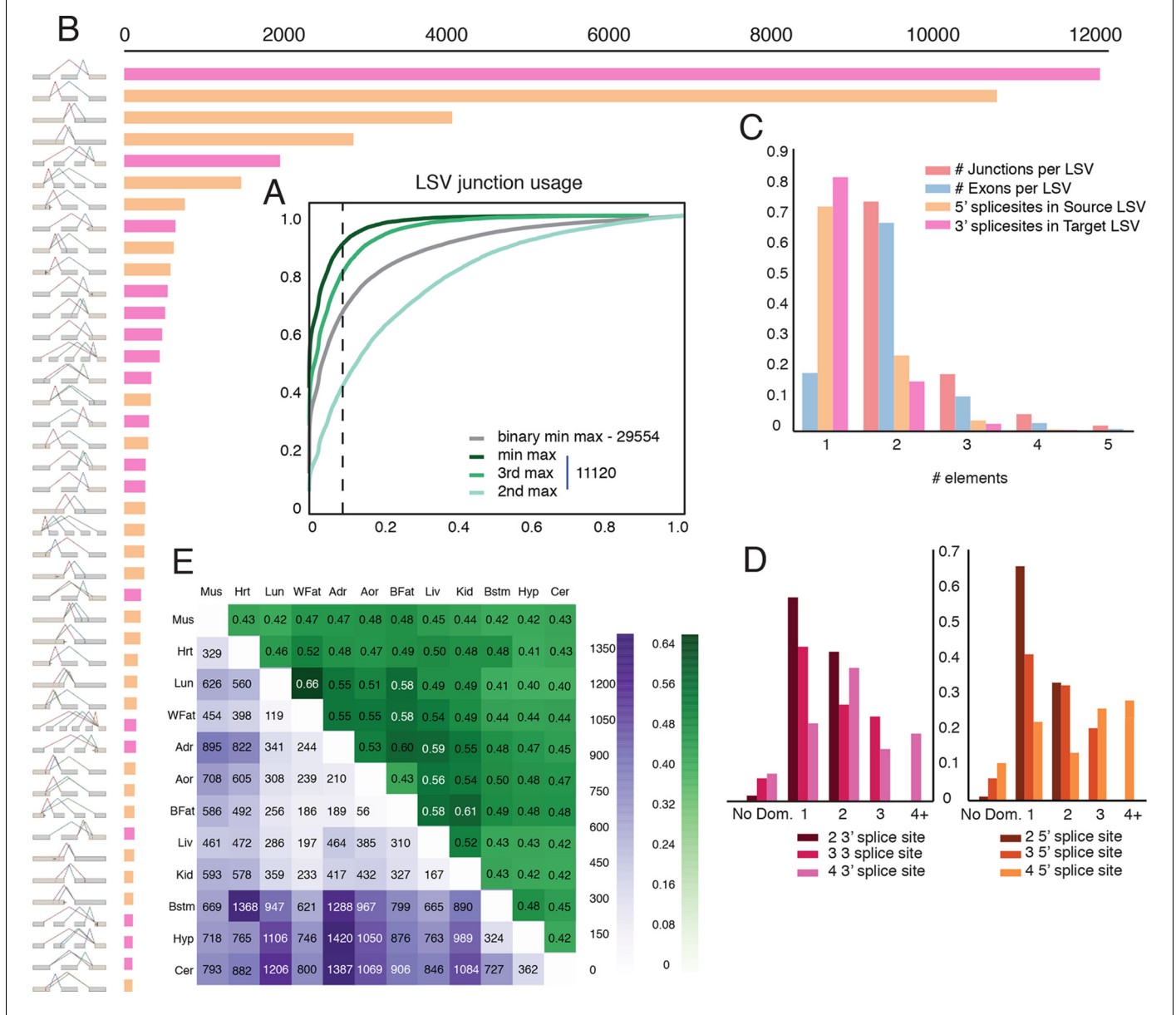

**Figure 4.** Genome wide view of exonic LSVs across twelve mouse tissues. (**A**) Cumulative distribution (CDF) for maximal junction inclusion (PSI) across tissues. Plot includes the least used junction in binary LSV (grey), the second, third and least used junction in complex LSVs (light, medium, dark green). Dashed vertical line denotes 10% inclusion. (**B**) Histogram of the most common exonic LSV types. (**C**) Histogram of the number of exons, junctions, 3' and 5' splice sites in all identified LSV. (**D**) Histogram of which 3' (left) or 5' (right) splice site are found to be dominant across all tissues and all LSVs. X-axis denotes the order of the splice site. Dominance is defined as $E[\Psi] > 0.6$. Cases with no dominant junction are represented by the bars on the far left. (**E**) The fraction of complex LSVs (green, top right) from the total number (purple, bottom left) of differentially spliced LSVs ($|E[\Delta\Psi]| > 0.2$) between pairs of tissues.

The following source data and figure supplement are available for figure 4:

**Source data 1.** dPSI values for all pairs of tissues.

**Figure supplement 1.** Intronic LSV detection and quantification.

'proximity rule', by which the splice site nearest to the reference exon is preferred (*Reed and Maniatis, 1986*), is commonly not reflected at the genomic level. Defining 'dominant' junctions as those included at least 60%, we found proximal junctions appear dominant in approximately two thirds of the cases involving binary LSVs (*Figure 4D*) while more complex LSV tend to have more evenly distributed inclusion levels with no dominant junction (*Figure 4D*, left bars). This more evenly distributed usage of exons and junctions in complex LSVs further supports possible functionality of multiple isoforms.

*Figure 4E* gives a genome wide view of the exonic LSVs that exhibit significant splicing changes (| E[ΔΨ]|> 20%) between mouse tissues. In line with previous reports (*Barash et al., 2010*; *Barbosa-Morais et al., 2012*), we find clear clusters for brain and muscle tissues (average of 875 and 657 changing LSVs, respectively), a weaker cluster for digestive tissues (liver, kidney) with an average of 501 changing LSVs, and lung as a unique signal (549 changing LSVs). Brain regions have a higher average of 927 (Cerebellum) to 840 (brainstem) changing LSVs compared to non-brain tissues. The number of LSVs changing between brain subregions varies between 36% and 57% of those changing between CNS and non-CNS tissues, with hypothalamus standing out as more similar to the two other CNS tissues (average of 937 and 343 changing LSVs when compared to non brain and other brain sub-regions, respectively). Overall, we find that complex LSVs make up almost 47% of the differentially spliced LSVs, a fold enrichment of 1.7 compared to their relative proportion of 27.5% in the quantifiable set ($P < 2.3 \times 10^{-278}$, binomial test).

## Complex LSV are enriched in regulated splicing that is associated with higher intronic conservation and specific protein features

Given the above result of complex LSV enrichment in tissue dependent splicing variations we decided to test whether this enrichment holds in other datasets that involve developmental stages, splice factor knockdowns, and disease. We performed a meta analysis of 31 mouse datasets that involve a total of 243 RNA-Seq experiments covering a variety of tissues, cell lines, developmental stages, and knockdowns of key splicing factors. To this set we also added a human dataset comparing Alzheimer's disease and healthy brain samples (*Figure 5A* and below). We found the median fraction of complex LSV in these datasets was 0.309 and their median fold enrichment in differentially spliced LSVs was 1.63, a significant enrichment in 30/32 of the datasets ($1.6 \times 10^{-322} <$ p-val $< 1 \times 10^{-3}$, Bonferroni corrected binomial test, see *Figure 5A*, and *Figure 5—source data1*). This consistent overrepresentation of complex LSVs among differentially spliced LSVs across a variety of contexts further suggests that complex LSVs are an important aspect of regulated alternative splicing.

Next, we asked how does the inclusion of junctions change across these datasets. For this, we took a conservative approach monitoring only the LSVs that have been already identified in normal tissues used to build the genome wide view of LSVs (*Figure 4*). *Figure 5B* shows over 20% of all complex LSVs detected in more than one sample had the third most differentially included junction exhibit |ΔΨ|> 10%, corresponding to 2,236 LSVs. Strikingly, these additional experimental contexts showed that over 39% of all complex LSVs detected in our normal tissue set had their third most included junction with Ψ > 10%, corresponding to 4,201 LSVs (*Figure 5—figure supplement 1*).

Finally, we plotted the conservation level around constitutive exons and differentially spliced LSVs shown in *Figure 4* that are either binary or complex (*Figure 5C*). Inline with previous reports, we found tissue regulated splicing involves significantly higher conservation in the intron proximal to the variable exonic segments, a region known to include *cis* elements to which tissue specific splice factors bind. However, we also found that differentially spliced complex LSVs exhibited significantly higher conservation levels in these regions compared to their binary counterparts. This finding may be the result of the more complex splicing changes that need to be controlled or tighter control associated with complex LSVs specific function. In summary, these different lines of evidence all support the functional relevance and utility of accurately mapping and quantifying complex splicing variations in genome wide studies.

The observed evolutionary pressure to conserve intronic segments around tissue dependent LSV raises the questions what are the functional consequences of LSVs and whether complex LSVs are functionally distinct from classical binary ones. To probe possible function we mapped exons in LSVs into their matching protein domains (see Material and methods). We then grouped LSV junctions based on whether they were part of binary or complex LSVs and whether they were differentially included across tissues. In line with previous works (*Ellis et al., 2012*), we find that binary LSVs, such

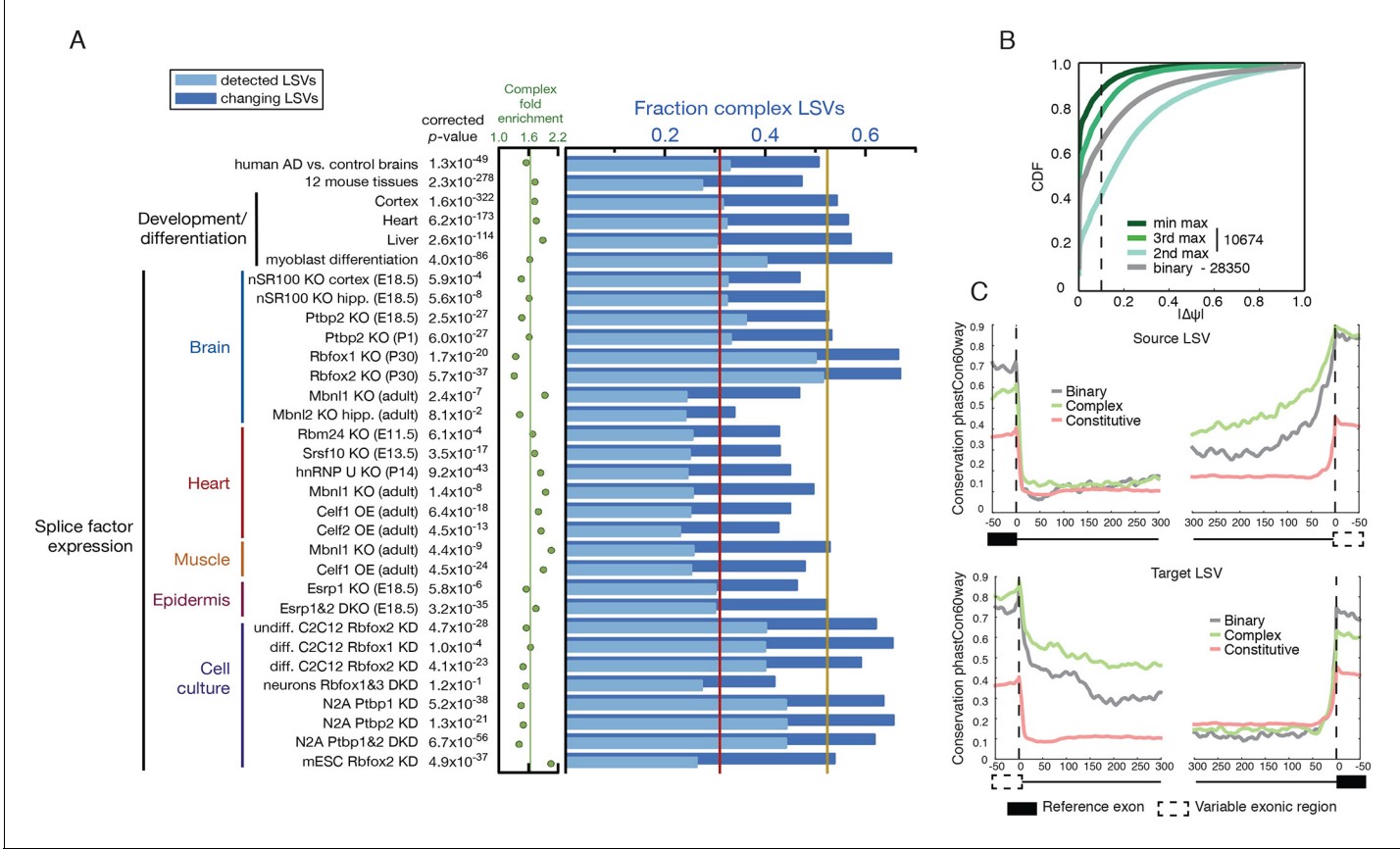

**Figure 5.** Meta analysis of complex LSVs. (A) Fold enrichment (green dots) of complex LSVs calculated by comparing the fraction of complex LSVs among differentially spliced LSVs (dark blue bars) to their relative proportion (light blue bars) in 32 datasets. The corrected p-value column on the left measures significance of the fold enrichment (binomial test, Bonferroni corrected *p*-value) Medians are displayed for fold enrichment (green line, 1.63), fraction of complex LSVs among changing LSVs (orange line, 0.52), and fraction of complex LSVs among all detected LSVs (red line, 0.31). Human AD versus healthy brain data corresponds to the cohort from (*Bai et al., 2013*). See *Figure 5—source data 1* for more information. (B) Empirical cumulative distribution function (CDF) of the maximal change of junction inclusion ( ΔΨ ) across all mouse datasets in *Figure 5A*. Only the LSVs detected in the twelve mouse tissues (*Figure 4*) are included. The plot includes junctions in binary LSVs (grey), and the second, third, and least changing junction in complex LSVs (light, medium, dark green). Dashed vertical line denotes ΔΨ of 10%. (C) Per nucleotide average conservation score (phastCons60 track) in regions proximal to single source (top) and single target (bottom) LSVs that were differentially spliced between any pair of tissues shown in *Figure 4*. The average is plotted for the subsets of complex (green) LSVs and binary (grey) LSVs as well as around a randomly selected set of constitutively spliced junctions (red, see Materials and methods for details).

The following source data and figure supplement are available for figure 5:

**Source data 1.** LSV enrichment meta analysis table.

**Figure supplement 1.** Empirical cumulative distribution function (CDF) of the maximal junction inclusion (E[Ψ]) across all mouse datasets in *Figure 5A*.

as cassette exons, which are also differentially included across tissues, more frequently affect low-complexity, disordered regions when compared to non-changing binary LSVs ($p<1\times10^{-4}$, corrected Fisher's exact test). Interestingly, differentially included complex LSVs are similarly enriched for such low-complexity regions ($p<1\times10^{-4}$), but also show enrichment for specific protein families (*e.g.* spectrin/filamin) and domains (e.g. RNA recognition motifs) when compared to non-changing complex LSVs. These families and domains are largely distinct from those enriched in binary LSVs (e.g. WW domains or coiled coils). The complete list of enriched protein features can be found in *Supplementary file 1*. Overall, this analysis suggests that regulated alternative splicing of both binary and complex LSVs can affect protein interactions via unstructured protein regions, or affect the inclusion of distinct protein domains in specific families.

## MAJIQ detects a novel, brain-specific, PTC-introducing, developmentally- regulated exon in *Ptbp1*

To further demonstrate the power of MAJIQ and our LSV based approach we validated a set of complex LSVs that exhibit tissue and brain region dependent splicing patterns. Surprisingly, this analysis revealed a previously uncharacterized, brain-specific exon in the gene encoding PTBP1, an extremely well studied splicing factor critical to neural development (*Keppetipola et al., 2012*) (*Figure 6A*, *Figure 6—figure supplement 1A*). While this novel exon remained undetected when running cufflinks (*Trapnell et al., 2010*) on this dataset (data not shown), expression of this novel exon as part of a complex LSV was supported by RT-PCR from cerebellum and adrenal tissues (*Figure 6B*, top) with good concordance with MAJIQ's PSI quantification (*Figure 6B*, bottom). Products including exon 14 were also weakly detected by RT-PCR of brainstem and hypothalamus-derived RNA, but not from any of the other eight tissues tested (*Figure 6—figure supplement 2*). Together these data strongly support exon 14 as brain-specific.

Interestingly, *Ptbp1* exon 14 shows conservation of splice sites between mouse and human and inserts multiple premature termination codons (PTCs) in both species, as well as in other mammals, before RMMs 3–4 of PTBP1 (*Figure 6—figure supplement 1A,B*), suggesting that mRNAs including this exon are likely targets of nonsense-mediated decay (NMD). Regulated alternative splicing that introduce PTCs is a common theme among numerous splicing factors (*Ni et al., 2007*) and exclusion of *Ptbp1* exon 16 (exon 11 in the literature) has already been identified and shown to induce NMD (*Figure 6—figure supplement 1A*) (*Wollerton et al., 2004*). Remarkably, exclusion of exon 16 is barely detectable in the brain regions examined and inclusion of exon 14 is not associated with this event (*Figure 6—figure supplement 1C*). Together, this suggests that these splicing events are independent mechanisms to control *Ptbp1* expression and that inclusion of novel exon 14 plays a larger role in the brain regions examined, with 26% of the *Ptbp1* transcripts in the cerebellum containing PTCs.

Embryonic down regulation of *Ptbp1* by miR-124 is crucial at the onset of neurogenesis (*Makeyev et al., 2007*) and leads a change in splicing programs (*Boutz et al., 2007*; *Keppetipola et al., 2012*), but cannot account for additional postnatal down regulation of this protein (*Boutz et al., 2007*; *Zheng et al., 2012*). Remarkably, MAJIQ analysis of RNA-seq data from mouse cortices across development (*Yan et al., 2015*) reveals clear developmental regulation of exon 14 with a dramatic increase in inclusion from P15 through adulthood (*Figure 6C*). Taken together, this complex LSV offers a novel mechanism for postnatal neuronal reduction in *Ptbp1*.

To identify putative regulators of novel exon 14, we used AVISPA (*Barash et al., 2013*), a web tool that utilizes splicing code models to suggest motifs important for tissue-specific splicing, and identified the [U]GCAUG binding motif of the Rbfox family as important for neuronal splicing outcome (*Figure 6D*). AVISPA's map of regulatory motifs pointed to a number of Rbfox binding sites downstream of exon 14 (*Figure 6A*). These motifs, perfectly conserved between mouse and human, suggested enhancement of inclusion by the Rbfox family (*Lovci et al., 2013*). Consistent with this regulatory hypothesis, MAJIQ analysis of RNA-seq data from one month old nestin-specific *Rbfox1* KO mice revealed a marked decrease in inclusion of exon 14 from ~16% in wild type mice to nearly undetectable in the KO (*Figure 6E*; *Figure 6—figure supplement 1D*) and similar decreased inclusion was observed upon *Rbfox2* KO (*Lovci et al., 2013*) (*Figure 6—figure supplement 1E*). Together these data demonstrate the power of MAJIQ, in combination with the VOILA and AVISPA analysis tools, in identifying previously uncharacterized isoforms and understanding the regulation of biologically important transcript variation.

## MAJIQ detects novel splicing variations in the CAMK2 family which are conserved, developmentally regulated, and dysregulated in AD

Several of the brain specific LSVs we detected were found in genes encoding calcium/calmodulin-dependent protein kinase II (CAMK2) subunits which regulate functions in the brain such as neurotransmitter synthesis and release, cellular transport, neurite extension, synaptic plasticity, learning and memory (*Griffith, 2004*). We focused on *Camk2d* and *Camk2g* as these exhibit complex changes and were expressed in nearly all tissues examined (*Figure 4—source data 1*). *Figure 1B* and *Figure 7—figure supplement 1B* show MAJIQ's analysis and matching RT-PCR validation of a *Camk2g* LSV containing three exons across five tissues. *Figure 7* shows similar verification for

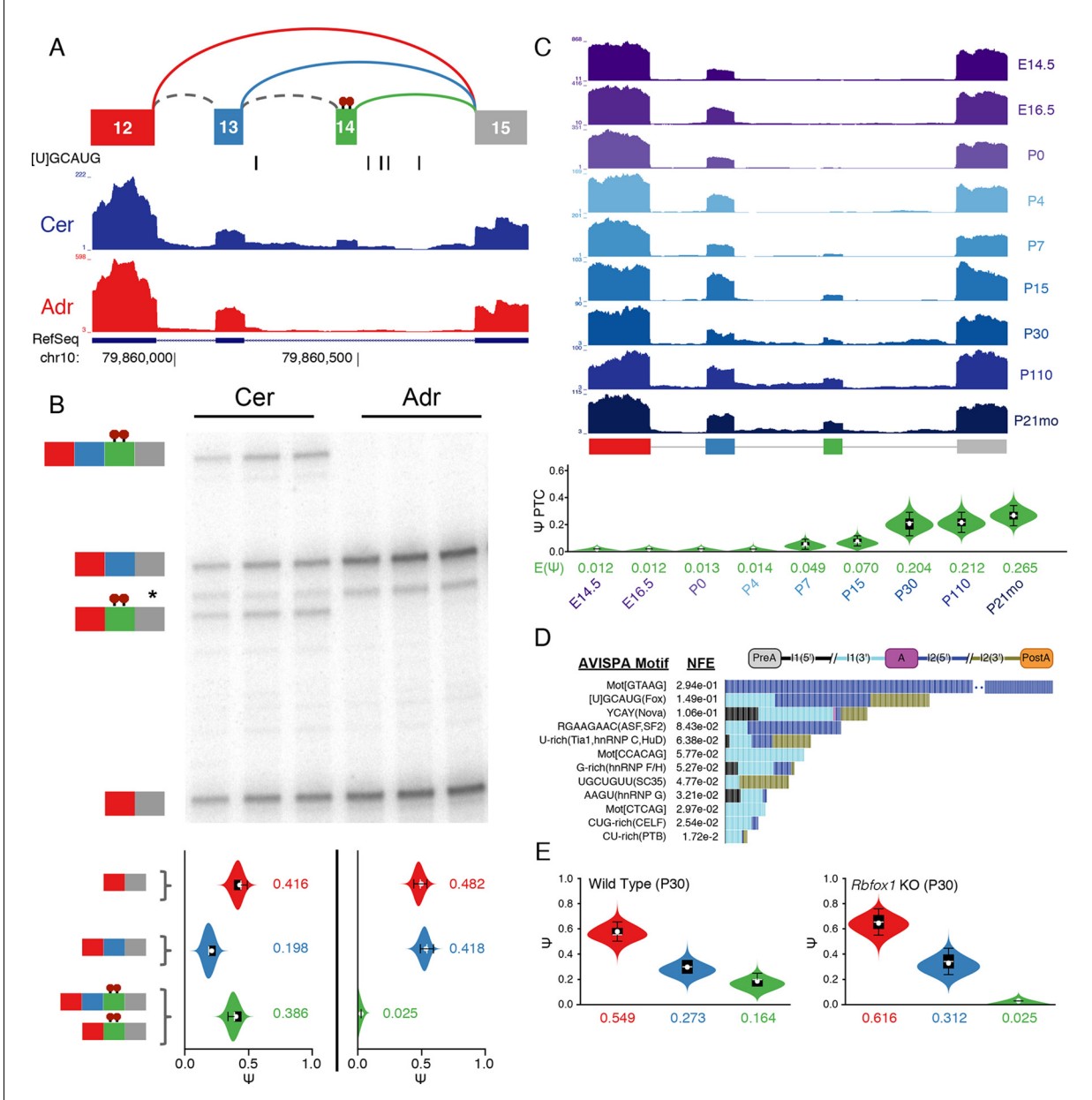

**Figure 6.** Identification of a novel, brain-specific, PTC-introducing, developmentally-regulated exon in *Ptbp1*. (A) Top: Splice graph representation of a complex target LSV containing a previously unannotated, PTC-introducing exon in *Ptbp1 (exon 14, green)*. Stop signs indicate multiple conserved premature termination codons. Bottom: UCSC Genome Browser tracks of RNA-seq reads from adrenal (red) and cerebellum (blue), and conserved Rbfox binding sites ([U]GCAUG) found within the bounds of this LSV. (B) Top panel: RT-PCR validation of RNA from replicate cerebellar and adrenal tissues with isoforms illustrated on the left. Asterisk denotes a background band that migrates non-specifically. Bottom panel: E[Ψ] violin plots of MAJIQ quantification for the colored junctions in (A). Matching isoforms are indicated on the left. (C) Top: RNA-seq reads from mouse cortices (*Yan et al., 2015*). Developmental time points indicated on the right with exons colored as in (A). Bottom: Ψ violin plots for the PTC-introducing exon 14 across brain development. (D) Top panel: Top regulatory motifs predicted by AVISPA to influence the neuronal-specific splicing of exon 14. Stacked bars represent the normalized feature effect (NFE) for each motif. Colors indicate the contribution of the corresponding motif in the region indicated in the inset. (E) MAJIQ Ψ quantification of the LSV shown in (A), using RNA-seq from one month old wild type whole brain (left) and nestin-specific *Rbfox1* KO littermates (right).

The following figure supplements are available for figure 6:

**Figure supplement 1.** Novel exon and PTCs in *Ptbp1* are conserved, independent from known PTC event, and regulated by Rbfox1 and 2.

*Figure 6 continued on next page*

Figure 6 continued

**Figure supplement 2.** RT-PCR validation of complex *Ptbp1* LSV across 11 mouse tissues.

another complex LSV but in *Camk2d*. In both cases, exon inclusion creates consensus NLS motifs (KKRK), which localize these subunits to the nucleus (*Braun and Schulman, 1995*). For *Camk2g* the NLS motif is contained in exon 15 whose inclusion levels are highest in the brain, particularly in the brainstem (*Figure 1B*, *Figure 7—figure supplement 1B*).

Several other important aspects of *Camk2d* splicing are accurately captured by MAJIQ. These include near 100% skipping of exons 21 through 23 in all non brain or muscle tissues (known in the literature as isoform C or Camk2δC, (*Xu et al., 2005*)), high relative inclusion of NLS containing exon 21 in heart (isoform B or Camk2δB), and high levels of isoform A (Camk2δA), which includes exons 22 and 23, in the brain regions examined (*Figure 7A*). This result is consistent with previous reports of *Camk2d* splicing patterns and isoform A being neuronal-specific (*Xu et al., 2005*). Importantly though, MAJIQ also detects isolated inclusion of exon 23 in the heart (*Figure 7A*, green junction), which is supported by both the RT-PCR experiment and analysis of an independent dataset across heart development (see below). Previous studies focused on splicing regulation of *Camk2d* in the heart used junction spanning primers that preclude detection of this highly utilized splicing choice (*Xu et al., 2005*; *Ye et al., 2015*).

Because CAMK2 has been implicated in neurodevelopment and is proposed to be critical for postnatal heart development (*Xu et al., 2005*), we next looked for developmental changes in LSVs by analyzing RNA-seq data derived from mouse cortices (*Yan et al., 2015*) and hearts (*Giudice et al., 2014*) at different time points. In the brain there is a switch in the splicing of *Camk2d* between the C and the A isoforms, reaching over 80% use of the A isoform by postnatal day 15, corresponding to a time of intense synaptogenesis and plasticity (*Licatalosi et al., 2012*) (*Figure 7B*, top). In the heart we see a more modest decrease in isoform C and increase in exon 23 only during postnatal heart development (*Figure 7B*, bottom, compare purple with green), consistent with results from RT-PCR from eight week old mice (*Figure 7A*). Notably, other CAMK2 subunits also displayed developmental dynamics in both tissues, such as inclusion of NLS containing exons in *Camk2g* and *Camk2a* (*Figure 7—figure supplement 1C* and *2*), an unannotated mouse cassette exon in *Camk2g* regulated by the Rbfox family (*Figure 7—figure supplement 1D–G*), and a complex LSV in the variable domain of *Camk2b* that affects autophosphorylation and is regulated by Ptbp2 (*Li et al., 2014*) (*Figure 7—figure supplement 3*).

Given the suggested role of calcium signaling in neurodegeneration (*Marambaud et al., 2009*) and CAMK2 implication in Alzheimer's disease (AD) (*Steiner et al., 1990*), we also analyzed RNA-seq data from three control brains and compared them to three AD brains (*Bai et al., 2013*). Strikingly, in *CAMK2D* we observe a marked decrease of ~38% of the neuronal specific isoform of the complex, developmentally-regulated mouse LSV we validated above, with reciprocal increase in the all exclusion, isoform C in AD brains (*Figure 7C*). We also observe changes in a *CAMK2G* LSV that corresponds to an unannotated mouse exon (*Figure 7—figure supplement 1D,E*). Importantly, these exons are perfectly conserved between mouse and human at the amino acid level, further suggesting physiologic importance of the novel splicing variations detected by MAJIQ. Finally, we validated that the observed CAMK2 splicing changes in AD brains can be reproduced in a second independent study. We used data from the AMP-AD Target Discovery Consortium (doi:10.7303/syn2580853) involving a larger cohort of 157 samples from AD patient's brains and 128 control samples, across three different brain sub regions (*Figure 7—figure supplement 4*). Overall, we detected approximately 200 LSVs that are reproducibly differentially spliced between AD and normal brains (see Methods) and enriched in GO terms such as cytoskeleton, GTPase regulator activity, and synapse organization (data not shown). This set constitutes approximately 12% of the changing LSVs detected in the original dataset, a fraction that grows to 21% but only 164 LSVs if stricter filtering is applied to both datasets (data not shown). This relatively low percentage of reproducible changes across the two datasets can be at least partially attributed to the small number of samples in the original study combined with an average of 1.8 fold lower coverage in the second, larger dataset. Notably though, among the reproducible set of differentially spliced LSVs 79 are complex, a significant, 1.2-fold enrichment compared to their relative proportion among all LSVs detected (p=0.04,

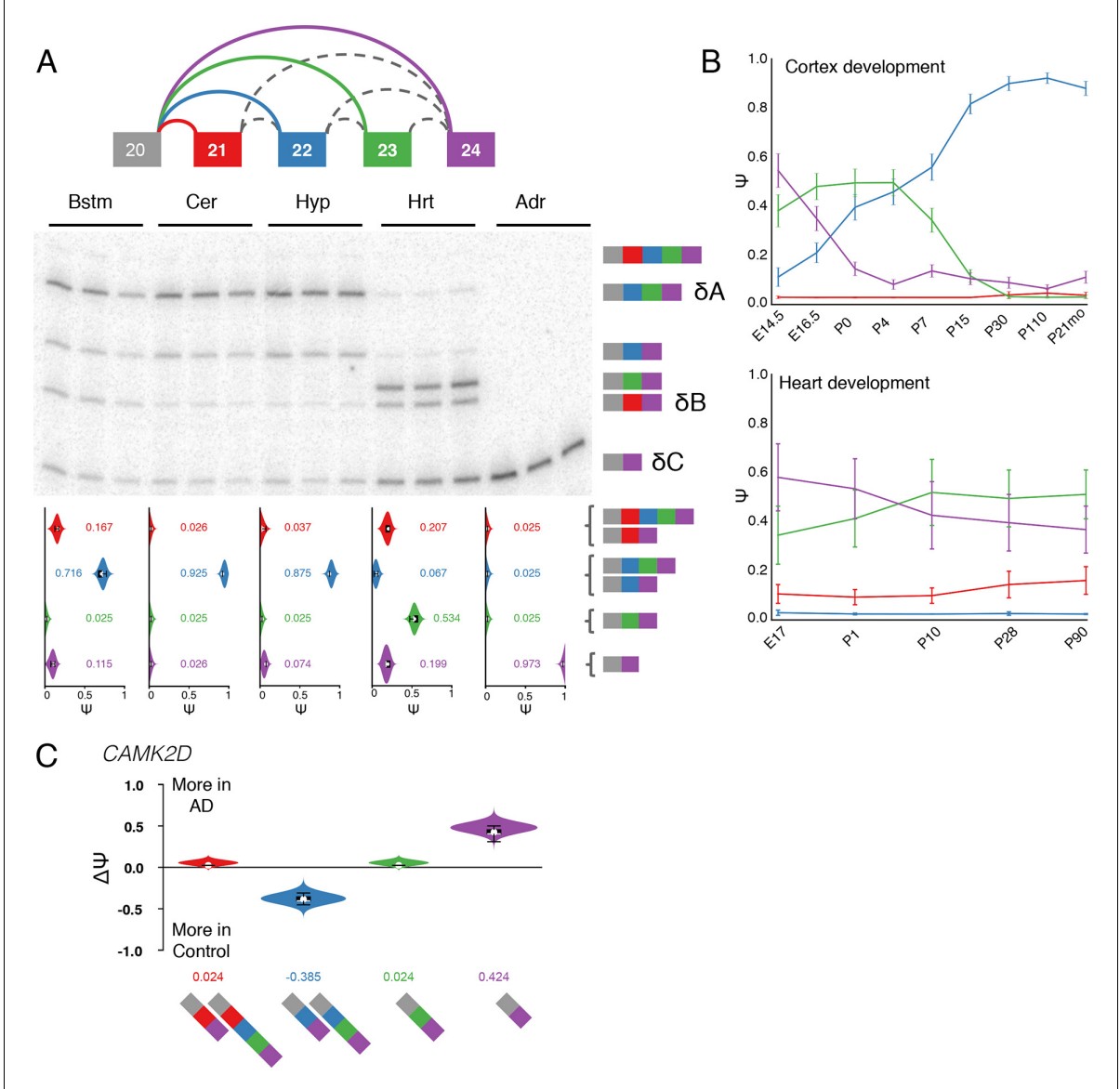

**Figure 7.** Camk2d LSV exhibits complex developmental dynamics and is misregulated in Alzheimer's disease. (**A**) Representation of complex source LSV in *Camk2d* with matching RT-PCR validation in five tissues (brainstem, cerebellum, hypothalamus, heart, and adrenal). Colored arcs represent the junctions quantified by MAJIQ for this LSV while dashed arcs correspond to junctions in the RNA-seq data that are not part of the quantified LSV. Violin plots on the bottom display Ψ quantifications (x-axis) for each of the colored junctions (y-axis) across the five tissues with appropriate isoforms from the gel on the right. Isoforms with known tissue-specific splicing patterns are labeled as in the literature (**B**) Line graphs of MAJIQ E[Ψ] quantification (y-axis) of junctions as in (**A**) across time points (x-axis) through cortex development (top) and heart development (bottom). Points represent mean Ψ and error bars represent one standard deviation in E[Ψ]. (**C**) ΔΨ quantification comparing changes between control and Alzheimer's patient brains of the homologous junctions illustrated in (**A**).

The following figure supplements are available for figure 7:

**Figure supplement 1.** Complex and de novo LSVs in Camk2g are developmentally regulated and dysregulated in Alzheimer's disease.

**Figure supplement 2.** LSV in Camk2a is developmentally regulated oppositely in the brain and heart.

**Figure supplement 3.** Developmentally controlled, complex LSV in *Camk2b* is regulated by Ptbp2.

**Figure supplement 4.** Analysis of CAMK2D, CAMK2D, and CLTA LSVs in an independent Alzheimer's cohort.

*Figure 7 continued*

**Figure supplement 5.** Complex alternative end of Alzheimer's-associated Klc1.

**Figure supplement 6.** Clta splicing is developmentally regulated and dysregulated in Alzheimer's Disease.

**Figure supplement 7.** Eif4g3 splicing shows brain subregion-specificity and a novel exon in muscle.

binomial test). While the validation and experimental follow up on these LSVs is beyond the scope of this paper these results and the related CAMK2 analysis demonstrate the usefulness of our combined approach for LSV detection, quantification, and visualization for disease studies.

Overall, our analysis of CAMK2 is in line with previous studies but also detects additional isoforms and exons that are conserved, developmentally regulated, and dysregulated in AD, making for a more accurate picture of CAMK2 splicing patterns. Additional complex LSVs we validated and analyzed include brain specific isoforms of the kinesin light chain *Klc1*, recently shown to be an amyloid-beta accumulation modifier (*Morihara et al., 2014*) (*Figure 7—figure supplement 5*); the clathrin light chain *Clta*, which displays developmental dynamics and dysregulation in both Alzheimer's disease cohorts (*Figure 7—figure supplement 6*, *Figure 7—figure supplement 4*); and the translation initiation factor scaffold *Eif4g3*, which has high inclusion of a cassette microexon specifically in cerebellum and a novel, muscle-specific exon (*Figure 7—figure supplement 7*).

## Discussion

The work presented here spans a wide spectrum of topics from a new formulation of transcriptome variations in units of local splicing variations (LSVs); through algorithms for detecting, quantification and visualization of LSVs; a genome wide map of LSVs; analysis of the prevalence and functional significance of complex LSVs; to validation of several complex LSVs that affect protein domains in developmentally regulated genes with key roles in neurogenesis or other brain functions. For the latter, we also demonstrated dysregulation in Alzheimer's disease using two independent datasets.

The new formulation of LSVs sheds light on what has thus far been mostly a 'dark side' of the transcriptome and RNA-Seq based studies, *i.e.* complex splicing variations. Several previous works aimed to address the apparent representational gap between full transcripts and the classical binary AS events. For example, (*Nagasaki et al., 2006*) developed an efficient bit array representation for the various exonic segments that make up different gene isoforms, and (*Sammeth et al., 2008*) suggested an elaborate notational system that allowed them to catalogue all the splicing variations in a given transcriptome, comparing the frequencies of different AS types across 12 metazoa. More recently, (*Pervouchine et al., 2013*) developed bam2ssj, a package implementing a general intron centric approach to estimate AS from RNA-Seq data that can capture non classical AS variations. bam2ssj gives a BAM-file–processing pipeline that counts junction reads to compute the ratio of inclusion levels either from the 5' or the 3' end of an intron, denoted $\Psi_5$ and $\Psi_3$. A different, graph based, approach was taken by (*Hu et al., 2013*) where a splice graph is divided into subunits termed alternative splicing modules (ASMs). ASMs are hierarchically structured, each capturing all the possible paths along a splice graph between specific start ('*single entry*') and end ('*single exit*') points. The matching algorithm, DiffSplice, then aims to identify cases of differential transcription of ASMs between two experimental conditions. All of these works differ substantially in the formulation of splicing variation, the underlying algorithms, and visualization approach, yet all share the effort to capture non classical AS types. In comparison, MAJIQ offers a unique approach that spans formulation, detection, quantification and visualization of splicing variations. Unlike ASMs, LSVs can be inferred directly from junction spanning reads and result in quantitative PSI and dPSI estimates, while MAJIQ's probabilistic model offers significant accuracy boost for PSI and dPSI estimates compared to alternative methods.

The importance of LSVs formulation is manifested in how common complex LSVs are in diverse metazoans, making up at least a third of observed LSVs in human and mouse. Complex LSVs are also enriched for regulated splicing when analyzing over thirty datasets across different tissues, developmental stages, splice factor knockdowns and neurodegenerative disease. In addition, LSV

formulation can be used to investigate substructures of the transcriptome. We found that the bio-chemically-based proximity rule is commonly overcome at the genomic level and that complex LSVs are less likely to have a dominant splice junction. As for LSVs possible function, our results indicate that tissue dependent binary and complex LSVs both tend to occur in unstructured regions known to affect protein-protein interactions, as well as in specific yet distinct protein domains and families.

In order to benefit from the new LSV formulation matching software is needed. The software we developed, MAJIQ, is LSV focused and compares favorably with available tools on AS quantification based both on RNA-Seq from biological replicates and on a compendium of over 200 RT-PCR experiments. Unlike many tools, MAJIQ supplements annotated transcriptomes with novel splice junctions, while VOILA allows the resulting LSVs to be interactively visualized within standard web browsers. Thus, MAJIQ and VOILA offer a compelling LSV centered addition to tools such as MISO (*Katz et al., 2010*), rMATS (*Shen et al., 2014*) and cuffdiff (*Trapnell et al., 2013*) that allow users to quantify whole isoforms relative abundance, alternative polyadenylation, or differential expression.

Immediate applications of the novel LSV framework and the MAJIQ software cover a wide spectrum. Examples include improved disease studies where transcriptome variations play a role, enhancing predictive models for splicing and for the effect of genetic variants, studying the regulatory underpinning of complex LSVs, and examining their evolutionary history. At the most basic level, our results illustrate the potential for novel discoveries in reanalyzing previously published data with the new LSV based methods. We anticipate the framework and resources provided here will form the basis of many additional new discoveries in diverse fields.

## Materials and methods

### RNA-Seq read mapping

All RNA-Seq was mapped using STAR (*Dobin et al., 2013*). STAR was run with alignSJoverhangMin 8. We created the STAR genome based on mm10 or hg19, with an in-house junction DB containing all possible junctions within each gene.

### LSV definition

An LSV (local splice variation) is defined as a split in a splice graph into or from a single exon, termed the reference exon. Single Source LSV (SS-LSV) correspond to splits from a reference exon to multiple 3' splice sites in downstream exons, single target LSV (ST-LSV) correspond to multiple 5' splice sites spliced to an upstream reference exon. The reference exon may include multiple 3' splice sites (ST-LSV) or 5' splice sites (SS-LSV). An LSV type is defined by the reference exon type (SS, ST) and the set of junctions it includes. Each junction is defined by the splice site ID in the reference exon, and the splice site ID in its target/source exon.

Under the above formulation some SS-LSV and ST-LSV may include exactly the same set of edges or one LSV may contain a subset of another LSV's edges. For example the SS-LSV from exon 4 and the ST-LSV into exon 5 in *Figure 1A* bottom are comprised of exactly the same edges, while the ST-LSV into exon 2 is a subset of the SS-LSV from exon 1. Such cases are easily detected and removed from further analysis to avoid redundancy.

It is important to note that under the LSV formulation classical cassette exons correspond to two distinct LSVs, a single source and a single target. These LSVs are not redundant as they correspond to different lines of experimental evidence; one from junction reads connecting the alternative middle exon with the upstream exon (SS-LSV) and one connecting the middle exon to the downstream exon (ST-LSV). The separate quantification for such LSVs, combined with the joint visualization using VOILA (see below), help distinguish between cases where the two LSVs give similar PSI or dPSI quantifications and cases where they disagree. A case of possible disagreement is illustrated in the last three exons of *Figure 1A*, where an alternative transcription start site and a third junction going into the last exon may lead to different PSI values.

### LSV as structural network motifs

The above definition gives a one-to-one mapping between a local splice graph split and an LSV type. Given a set of LSVs we can compute a distribution over their types or group several types together to detect a distribution over specific LSV features. We note that unlike the analysis of

network motifs in *Milo et al. (2002)*, we do not compare LSVs to random connections in a network as the null hypothesis, but rather to a sequential network where all exons are connected via a single path. Thus, we compute a distribution over relevant statistics such as the number of junctions in the reference exon, the total number of junctions or the total number of exons in the LSV (*Figure 4*).

## MAJIQ

MAJIQ is comprised of two main components, a builder and a quantifier. The builder analyzes a given set of RNA-Seq experiments and a transcriptome database to detect LSVs (either known or de-novo) and create a splice graph for each gene in the database. The quantifier subsequently estimates PSI or dPSI for LSVs detected by the builder.

### MAJIQ Builder

The builder accepts as input a list of RNA-Seq indexed BAM files and a transcriptome database. For each gene defined in the database it determines all its known exons, associated 3'/5' splice sites, and the splice graph edges (i.e. splice sites spliced together). It then scans the BAM files to find which of those edges are supported by RNA-Seq reads, and which de-novo splice sites and edges should be added based on junction reads. A user-controlled filter defines which edge is considered 'reliable' to be included in the splice graph. The default is set to "appears in the database or has at least two reads from two different positions". In order to create the splice graph, MAJIQ combines the known transcriptome with the reliable de-novo junctions. If de-novo 3' or 5' splice sites are found outside the boundaries of any annotated exons, they are connected to the proximal upstream or downstream exon, respectively. If such a de-novo 3' splice site is followed by a de-novo 5' splice site then the area in between is denoted as a putative de-novo exon. However, de-novo exonic regions added to the splice graph are not allowed to exceed a user-defined threshold. The default threshold is set to 500bp, which corresponds to approximately the 95 percentile of known exons length in the mouse genome. In cases where a de-novo exonic extension might have exceeded this threshold it is instead used to create an 'open ended' putative de-novo exonic region with its boundaries marked accordingly. All the de-novo junctions and exonic regions are marked in green in VOILA's output (see for example *Figure 6—figure supplement 1A*). Finally, for retained introns, an additional filter is added for the average coverage in consecutive windows across the intron. The default is set to 1.5 (see Supplementary *Figure 4—figure supplement 1A, B*) but it is important to adjust this threshold depending on the coverage depth and how permissive one wants to be when calling retained introns.

In the next step, the builder creates a list of LSVs that are considered to 'exist in the data' based on a second user defined filter. The default for this filter is 'all the LSV junctions are reliable and at least one junction has at least two reads from at least two positions'. Intuitively this setting retains only LSVs for which there is evidence for expression of at least some gene isoforms involved in that LSV though not all edges involved in the LSV will necessarily be found in the data. The MAJIQ Builder outputs two types of binary files, the first including the LSVs to be quantified by the MAJIQ quantifier, the second including the splice graphs to be visualized by VOILA.

### MAJIQ quantifier

The quantifier estimates the fraction each LSV junction is selected, denoted percent selected index (PSI or $\Psi$), or the changes in each junction's PSI between two experimental conditions with or without replicates (dPSI or $\Delta\Psi$). These fractions are inferred from short sequence reads that span across junctions (junction reads), whose distribution can be affected by many factors. Consequently, MAJIQ quantifies PSI and dPSI not as a point estimate but as a posterior distribution over possible fractions in the range [0,1]. Importantly, the quantifier screens the builder's list of LSVs for those that are deemed 'quantifiable' by a user-defined filter. Intuitively, the sparsity of RNA-Seq data results in many LSVs being reliably detected by RNA-Seq yet lacking sufficient read coverage to accurately quantify PSI or to confidently infer significant changes in PSI across conditions. By filtering those from downstream analysis by the quantifier significant computational resources can be saved. The default for the quantifiable filter is at least 10 reads from at least 3 positions.

The first step in MAJIQ's quantification is estimating the read rate per position in each junction, after correcting for GC content bias (*Risso et al., 2011*). Note that here the position's read rate

corresponds to reads that start/end at that position, not reads that overlap it. The estimation of each junction's read rate involves three components: A global parametric model per experiment for read count variability; a stack removal procedure; and a local estimator for read rate derived from bootstrapping over each junction's relevant positions. For the global parametric model MAJIQ uses the zero truncated negative binomial (ZTNB) distribution. The dispersion parameter $r_t$ is optimized per experiment $t$ by bootstrapping over non-zero positions in a randomly selected set of up to 10,000 quantifiable junctions. Next, given the derived negative binomial model with dispersion $r$, MAJIQ performs a screening step to detect possible read stacks. A read stack is defined as a position i in junction j with an observed read rate $x_{j,i}$ which is highly unlikely given $r$ and the average read rate in other non zero positions in the junction. After experimentation with the effect on reproducibility, a conservative threshold of p-val $\leq 10^{-7}$ was set as the default to flag possible read stacks. Flagged positions and their respective reads are then removed from further consideration. Finally, we noticed that even after fitting a global dispersion parameter per experiment $r_t$ and discarding read stacks the data still exhibits variability not fully accounted for by this model (data not shown). Therefore, and in order to account for local dispersion (i.e. at a specific junction) we bootstrap N positions from the relevant set of positions to get an estimate for the read rate in junction $j$

$$\mu^j = W_j \frac{1}{N} \sum_{n=1}^{N} c^{j,i_n}$$ where $c^{j,i_n}$ is the observed number of reads that start in the $i_n$ sampled position

and $W_j$ is the number of relevant positions in junction $j$. Here, the relevant positions refer to those where uniquely mapped, non-stack, reads start. By repeating this procedure M times we get an empirical distribution over $\mu^j$ estimates. These M samples are then used for computing posterior distributions over PSI and ΔPSI.

Estimating the percent selected index (PSI, or Ψ) per junction $j$ in a given LSV $e$ in experiment $t$ requires to derive a posterior distribution over multinomial distributions $\Psi_e = \{\Psi_{e,j}\}^J$, s.t. $\sum \Psi_{e,j} = 1$, $\forall e,j\ 0 \leq \Psi_{e,j} \leq 1$. Previous works concentrated on common cases involving two junctions such as cassette exons, where the posterior for Ψ can be computed in closed form using for example a Beta prior. For more complex cases where $J > 2$ the PSI posterior was commonly computed either as a point estimate (e.g., ML estimator using EM) or using MCMC sampling techniques (**Katz et al., 2010**). In general, sampling based estimation for $\Psi_{e,j}$ or $\Delta\Psi_{e,j}$ (below) scales exponentially in the number of junctions J and is also hard to visualize beyond $J = 2$. However, noting that in most cases researchers are interested in relative abundance of specific variants rather than a complete distribution over all isoforms, MAJIQ side steps these issues by computing only the posterior marginal distributions per variant. This computation scales linearly with $J$ and simplifies both downstream analysis and visualization of the results. It has been previously observed that alternative junctions in a given experimental condition generally tend to be either highly included or highly excluded (**Shen et al., 2012**; **Wu et al., 2011**). In line with these observations and based on fitting empirical distributions of observed PSI (data not shown) MAJIQ uses the following prior: $P_0(\Psi_{e,j}) \sim Beta(\alpha = \frac{1}{J}\eta, \beta = \frac{J-1}{J}\eta)$. The default is $\eta = 1$ resulting in a Jeffery prior that encourages either high inclusion or exclusion levels, but any (α, β) can be set. MAJIQ then uses the M read samples per junction (see above) to get a posterior $\Psi_{e,j}$ as an average over those posterior distributions.

PSI and ΔPSI are modeled as continuous random variables confined to the intervals [0, 1] and [−1, 1] respectively. In practice though, the required precision for these quantities is limited by both the problem being studied and the experimental techniques used to validate results (e.g., RT-PCR). This observation motivates MAJIQ's discretized representation of possible PSI and ΔPSI. The discretization level, controlled by a tunable resolution parameter V, allows explicit tradeoff between accuracy and computation cost. Setting V = 40 (default) results in 2.5% PSI resolution. This discretization allows MAJIQ's implementation to maintain and visualize a full distribution over PSI and ΔPSI, exploit efficient matrix operations for the entire range of Ψ values and avoid costly sampling procedures.

MAJIQ's estimation of $\Delta\Psi^e_{t,t'}$ for LSV $e$ between experiments $t,t'$ is based on a joint prior $P_0(\Psi_t, \Psi_{t'})$. While many previous works implicitly assume independence (i.e., $P_0(\Psi_t, \Psi_{t'}) \sim P_0(\Psi_t) P_0(\Psi_{t'})$) both MAJIQ and rMATS (**Shen et al., 2014**) use a prior biased towards similar $\Psi t, \Psi t'$ values, which helps overcome falsely reporting high ΔΨ due to fluctuations in small read counts. However, unlike rMATS that uses a multivariate uniform prior for ΔΨ, MAJIQ combines the $P_0(\Psi)$ prior described above with a ΔΨ prior: $P_0(\Psi_t, \Psi_{t'}) = P_0(\Psi_t)P_0(\Psi_{t'})P_0(\Delta\Psi = \Psi_t - \Psi_{t'})$, with $P_0(\Delta\Psi)$ having the form of a mixture of beta distributions:

$$P_0(\Delta\Psi) = \sum_{K=1}^{K} P(k)Beta(\Delta\Psi|\alpha(k),\beta(k)).$$

After some experimentation and measuring the effect on LSV quantification (data not shown), we found the following settings worked well. We set K = 3 with one component set as a spike at $\Delta\Psi$ = 0, the second as a beta distribution of small perturbations around 0, and the third mixture component set to a flat uniform prior ($\alpha = \beta = 1$). Given the above prior, the joint posterior distribution is given by:

$$P(\Psi_t^e, \Psi_{t'}^e|D_t^e, D_{t'}^e) \propto P_0(\Delta\Psi_{t,t'}^e)P(\Psi_t^e|D_t^e)P(\Psi_{t'}^e|D_{t'}^e),$$

where $D_t^e$ represents the set of estimated number of reads per junction in the m-th sample (see above) and $P(\Psi_t^e|D_t^e)$ is the posterior beta distribution given the observed reads. Similarly, when comparing two conditions $T$, $T'$ with replicates we have:

$$P(\Psi_T^e, \Psi_{T'}^e|D_T^e, D_{T'}^e) \propto P_0(\Delta\Psi_{t,t'}^e)P(\Psi_T^e|D_1^e,\ldots,D_T^e)P(\Psi_T^e|D_1^e,\ldots,D_T^e)$$

with the last two terms decomposing elegantly by the chain rule for the conjugate beta prior. More information regarding MAJIQ's usage and parameters can be found in the software's tutorial, available at majiq.biociphers.org.

## VOILA

VOILA creates HTML5 based visualization of gene splice graphs, LSVs, PSI and dPSI estimates. It uses two types of input files: a binary file output from MAJIQ builder summarizing gene splice graphs, and another binary file from MAJIQ quantifier summarizing LSV PSI/dPSI quantifications. The HTML5 lists splice graphs and associated LSVs according to user defined filters. Distributions over PSI or dPSI are represented using violin plots and each splice graph and LSV is also linked to the UCSC genome browser to allow comparison to raw reads or other track information. Interactive filters allow users to select which types of LSVs to display while a table view allows users to sort and search LSVs.

The VOILA splice graphs, LSVs cartoons and violin plots are shown in *Figure 6,7* and their respective supplementary figures. The original VOILA plots used for these figures can be found at: majiq. biociphers.org. More information regarding VOILA usage and parameters can be found in the software's user guide, available at majiq.biociphers.org.

## PSI reproducibility

PSI reproducibility by RNA-Seq from biological replicates was evaluated using the following procedure. First, MAJIQ Builder was executed to detect the union set of LSVs in a set of biological replicates of hippocampus and liver from *Keane et al. (2011)*. To avoid redundancy and enable comparison to other methods only a single junction from binary LSVs were included in downstream analysis. Next, for each replicate pair the difference in LSV quantification for each LSV was computed as $R(\Psi^{MAJIQ}) = E[\Psi_{r1}]-E[\Psi_{r1}]$. LSVs that were only detected in one of the replicates were discarded. The same set of LSVs were fed into MISO using the MAJIQ Builder GFF3 output file and the same procedure was executed to compute $R(\Psi^{MISO})$. This procedure was repeated 6 times to compute the mean and standard errors for the empirical $R(\Psi)$ PDF shown in *Figure 2—figure supplement 1C*. The empirical PDF and standard error for the difference in reproducibility $\Delta R = R^{MISO}- R^{MAJIQ}$ (*Figure 2—figure supplement 1C* inset graph) were computed by a similar procedure.

PSI reproducibility by RT-PCR was evaluated using the following procedure. For the data from *Zhang et al. (2014)*, we first selected LSVs that were estimated by MAJIQ to be differentially spliced with high confidence (P( $\Delta\Psi$ >0.2) > 0.95)) when using three samples from cerebellum and liver. This allowed us to also assess dPSI reproducibility for a wide range of dPSI values (see below). Next, for each LSV the total number of reads starting at positions within all the LSV's junctions in each replicate were summed together for each tissue. Then, the LSVs were binned by the average total read coverage in the two tissues. Bins were defined to be: 10–30, 30–40, 40–80, 80–200, and above 200 reads. From each such bin, a set of LSVs was randomly selected for RT-PCR validation. Each RT-PCR was executed in triplicates (see below). Finally, the average PSI by RT-PCR and the expected PSI by either MAJIQ or MISO were used to produce *Figure 2*, and *Figure 2—figure supplement 1*.

MISO was executed with default parameters. For the stimulated and unstimulated T-Cell dataset, we collected a compendium of historical RT-PCR quantifications for previously annotated cassette exons. These experiments were executed by different Lynch lab members across several years and pre selected for specific studies regardless of dPSI or RNA-Seq coverage level. The vast majority of these cassette exons did not exhibit differential splicing between stimulated and unstimulated cells and some lacked triplicates. This set of previously annotated cassette exons was mapped to MAJIQ's LVS and then quantified using RNA-Seq from *Cole et al. (2015)* (*Figure 2*, *Figure 2—figure supplement 1*, circle shaped points).

## dPSI reproducibility

dPSI reproducibility by RNA-Seq from biological replicates was evaluated using the following procedure. First, the MAJIQ Builder was executed for all replicates of hippocampus and liver experiments from *Keane et al. (2011)*, yielding the union of all LSVs in these experiments. Next, for each liver and hippocampus pair of experiments, all quantifiable LSVs were ranked according to their $E[\Delta\Psi]$ and the set of *N* LSVs with significant splicing changes at high confidence was defined as LSVs for which $P(\Delta\Psi > 0.2) > 0.95$. This threshold was selected to be conservative, but see *Figure 2—figure supplement 2A* for more relaxed thresholds. This process was then repeated in another pair of experiments and the relative rank of the original set of N LSVs was recorded. The reproducibility ratio $RR(\frac{n}{N})$ of any ranked LSVs subset $n \in 1 \ldots N$ was defined as the fraction $\frac{n*}{N}$ where *n\** is the subset of the first n ranked LSVs that were in the N best ranked LSVs by the replicate experiments. Similar to the IRD statistic used to assess reproducibility of Chip-Seq peak calling (*Li et al., 2011*), a perfect RR(n) graph follows the diagonal line. Unlike IRD though, the RR statistic is invariant to small or even complete perturbations in the relative rank of the top ranked LSVs. Intuitively, this means that the RR will remain the same as long as the same subset *n\** makes the best *N* cutoff. It is important to note that the RR value can vary greatly, affected by biological, experimental, and technical factors. Nonetheless, one can use the RR to assess reproducibility in specific settings, or compare dPSI reproducibility by different algorithms under the same experimental setup.

An inherent challenge in comparing MAJIQ to other methods is that MAJIQ quantifies LSVs while other methods quantify the classical AS event types. One complication as a result of that is that while redundant LSVs are removed (see above) different LSVs may still partially overlap. A good example for that are cassette exons. In the LSV formulation a differentially included cassette exon may have two LSVs that support it, corresponding to different lines of experimental evidence (junction reads from the up and downstream exons) but other methods/tools will only count this exon as a single event. This in turn may bias both the reproducibility ratio (RR) and detection power (N) in favor of MAJIQ. In our experiments, when we ignored such possible overlap of LSVs the reproducibility ratio remained the same but the number of differentially spliced LSVs detected was significantly higher (RR=86%, N = 752, data not shown). In order to avoid such a bias in favor of MAJIQ we implemented a conservative approach where the ranked LSVs are filtered so that no LSV contained overlapping exons with another LSV. We note this is a conservative filter as there may be complex LSVs that involve multiple differentially spliced exons, or cases where the same exone involves different variations (*e.g.* skipping the exon but also alternative 3' or 5' splice sites). In such cases only a single LSV would pass that filter while the methods we compared to would still be able to retain separate AS events for those.

dPSI reproducibility for MISO was evaluated by the following procedure. First, we followed MISO's (*Katz et al., 2010*) guidelines for performing exon-centeric analysis (i.e. AS events) rather than whole transcripts analysis. For this, we used the set of alternative events for the mm10 mouse genome provided by MISO. We indexed the GFF3 file and ran MISO with default parameters on the same data pairs of experiments described above to compute expected dPSI ($E[\Delta\Psi^{MISO}]$). Finally, we ranked LSVs by decreasing expected dPSI and computed the reproducibility ratio (RR) as described above. As MISO does not supply a statistical criteria for selecting the number of events (N) from its ranked list, we used the number produced by rMATS. Changing N to the higher number of LSVs detected by MAJIQ degraded MISO's performance (data not shown).

dPSI reproducibility for rMATS was evaluated by the following procedure. We ran rMATS (*Shen et al., 2014*) with replicates (groups) and without them (pairs), using ENSEMBL annotation file in GTF format. rMATS estimates differential expression for each one of the classic alternative splicing

events it identifies from the annotation file (exon skipping, 5 and 3 prime splice site donor/acceptor, mutually exclusive exons and intron retention). We used the default parameters except for the cutoff employed to compute the FDR associated to each AS event quantification, which was set to 0.2 (see *Figure 2—figure supplement 2A* for the impact in reproducibility of different cutoffs). Lastly, we extracted the RR for confident changing AS events identified by rMATS (FDR < 0.05, P( ΔΨ >0.2) as reported in the rMATS output file).

dPSI reproducibility for the Naive Bootstrapping approach used in *Xiong et al. (2015)* for cassette exons was adopted for LSVs using the following procedure. First, we implemented the bootstrapping over junction positions described in *Xiong et al. (2015)*, with the same beta prior to avoid zero read counts. These samples gave an empirical distribution over possible PSI values and these were subsequently used to estimate the expected PSI. Similar to MISO, the Naive Bootstrapping approach does not assume a joint prior so that the expected dPSI estimates are simply the difference in the expected PSI in each experiment. The resulting expected dPSI was then used to rank the LSVs, filter them for possible overlap of exons, and compute RR as described above.

dPSI accuracy by RT-PCR was evaluated by the same procedure as that described above for PSI. $\Delta\Psi^{RT}$ was then computed as the difference between the average of each triplicate set of experiments in cerebellum and liver or the difference between previously recorded measurements in the Lynch Lab for the stimulated vs. unstimulated T-Cells. dPSI reproducibility by RT-PCR was defined as cases for which $\Delta\Psi^{RT}$ >20%. This definition allowed assessing false positives and false negatives (*Figure 2—figure supplement 1B*, *Figure 2—figure supplement 2B*).

## Protein feature enrichment

In order to construct the LSV junctions and protein features (PF) table we first built the union set of LSVs detected from *Zhang et al. (2014)*. We used ENSEMBL RESTful services [http://www.ncbi.nlm.nih.gov/pubmed/25236461] to retrieve PF along with their genomic coordinates associated with transcripts containing LSVs. Next, we annotated each LSV junction by PF that overlap its reference exon and its target/source exon, discarding junctions in non-coding regions. Because the overlap of a LSV junction region and a protein feature can be partial, we considered a PF to be associated with a junction when there was at least a 20% overlap. Lastly, we labeled as changing junctions those that had an estimated delta psi greater than 20% in any two tissues.

We assessed relative enrichment of PFs using the following procedure when comparing groups of LSV junctions such as changing vs. unchanging, or binary vs. complex. For each PF we computed the p-value by Fisher's Exact Test (FET) for its distribution between the two junction groups compared. To correct for multiple hypotheses testing while accounting for the high correlation between some PF we applied a permutation based testing procedure (Column M). Specifically, we shuffled the labels (e.g. changing, unchanging) but controlled for the LSV origin of each junction. Thus, junctions from the same LSV were randomly switched with junctions from an LSV with the same number of junctions. This procedure guarantees that the number of labeled junctions remains the same, helps control for correlation between PF of junctions in the same LSV and for the distribution of LSV types. We repeated this process 10000 times and then calculate an empirical corrected FET p-value. The results from this analysis are included in *Supplementary file 1*.

## LSVs species analysis

RNA-Seq data for lizard and chicken was downloaded from *Barbosa-Morais et al. (2012)*; opossum and chimp datasets were downloaded from *Brawand et al. (2011)*. RNA-Seq for human was downloaded from Illumina's Body Atlas 2.0 (NCBI GSE30611). Transcriptomes were downloaded from Ensembl for lizard (genome assembly AnoCar2.0), chicken (assembly Galgal4), opossum (assembly monDom5), chimp (assembly Pan_troglodytes-2.1.3), mouse (assembly GRCm38.p3) and human (assembly GRCh38.p2). For mouse and human RefSeq transcriptome annotations were used for comparison (*Figure 3*). The latest genome builds annotated in RefSeq were used, GRCm38/mm10 for mouse and GRCh37/hg19 for human.

## Meta analysis of complex LSVs across datasets

In order to assess the prevalence and potential enrichment of complex LSVs across additional datasets beyond the 12 mouse tissues, we analyzed a number of additional datasets shown in *Figure 5A*

and *Figure 5—source data 1*. All raw data was downloaded from SRA and mapped using STAR as described above. In a select number of older or low-coverage experiments, mapped reads from replicates were pooled together before analyzing with MAJIQ (see 'Notes on processing', *Figure 5—source data 1*). MAJIQ dPSI was run for each comparison (*e.g.*, tissue pairs, pairwise developmental time points, control versus altered splice factor expression). LSVs were considered differentially spliced if E[ΔΨ] > 20%. For datasets with multiple conditions (e.g. 12 tissues, or multiple developmental timepoints), the union of differentially spliced LSVs and all detected LSVs between all pairwise comparisons was considered.

The enrichment of complex LSVs in the differentially spliced group compared to their relative proportion among all detected LSVs in each dataset was evaluated using a binomial test, with a Bonferroni correction for the number of datasets used. All counts, SRA and GEO data accession numbers, and PubMed IDs for each study are detailed in *Figure 5—source data 1*.

To assess the distributions of PSI and dPSI across all datasets in *Figure 5B* and *Figure 5—figure supplement 1*, we considered only the LSVs and junctions detected across the 12 mouse tissues and required exact matches to these junctions in the additional datasets in order to consider those PSI or dPSI values in the analysis. This conservative approach ensured we only monitored 'natural' LSVs and no LSVs that are unique to a specific cell line or KD experiment.

## Analysis of splicing changes in Alzheimer's disease in two cohorts

In order to validate splicing changes in AD identified for the complex LSVs examined in this study (*CAMK2D, CAMK2G*, and *CLTA*) we took all differentially spliced LSVs we identified from the 3 healthy and 3 AD brains (*Bai et al., 2013*) and looked for similar changes in an independent, larger cohort. We used data from the Mount Sinai Brain Bank (MSBB) RNA sequencing study (ID: syn3157743, accessed at https://www.synapse.org/#!Synapse:syn3157743)

We focused on samples that came from healthy brains and definite AD brains, based on CERAD Neuropathology Criteria given, across the following brain regions: frontal pole (healthy: n=58, AD: n=62); superior temporal gyrus (healthy: n=37, AD: n=50); parahippocampal gyrus (healthy: n=33, AD: n=45). Because overall coverage was lower in these datasets compared to the original cohort, which affects the ability to detect intron retention (data not shown), we ran MAJIQ Builder on both datasets with a high threshold for IR detection (–min_intronic_cov 1000) in order to only compare exonic LSVs. Additionally, to account for heterogeneity in the data and to save computational time we considered PSI values for each patient separately, as opposed to running all possible pairwise dPSI comparisons.

An LSV that was changing in the first cohort was considered validated if in the MSBB cohort the distribution of PSI values for the most changing junction was significantly different between healthy and AD individuals in at least one brain subregion (p<0.05, two-tailed rank sum test) with a difference in the median PSI of ≥ 10% in the same direction as the original cohort. This lead to 199 LSVs in 145 genes changing in both cohorts. Finally, DAVID was used to find enriched GO terms among these genes with shared differentially spliced LSVs between the two cohorts using default parameters (*Huang et al., 2008*).

## LSV conservation analysis

The conservation plots in *Figure 5C* were generated using the union of the changing LSV from the 66 pairwise tissue comparisons shown in *Figure 4E*. For each such LSV we extracted the phastCons60 conservation scores for vertebrates (*Siepel et al., 2005*) for the first 50 positions in each exon and the first 300 intronic positions proximal to each exon.

It is not immediately clear which of the variable regions in a complex LSV (left hand plot for the single source LSVs, right hand plot for the single target LSVs in *Figure 5C*) should be included in such conservation analysis. Previous work focused on binary cases of cassette exons and compared constitutive exons to the regions around the alternative exon, which tend to be more conserved. Since the focus of this analysis was on conservation of possible regulation in LSV units we chose to apply the max function for each position in such variable LSV regions. To partially correct for the possible bias for high scores that the max operation may introduce we also applied it to the binary LSVs and to randomly selected sets of K constitutive exons, where the size K is sampled based on the distribution of number of exons in LSVs (*Figure 4C*). Overall we sampled 5000 such sets for the

constitutive regions plot. Finally, the lines in *Figure 5C* were smoothed using a 5 bases sliding window.

## RT-PCR validations

Total RNA was extracted from mouse tissues as described previously (*Zhang et al., 2014*). For each tissue three samples corresponding to RNA from circadian times 31, 41, and 53 were used for validation. For additional validations we used total RNA extracted from a clonal Jurkat T cell line (JSL1, described in detail previously [*Lynch and Weiss, 2000*]) cultured in RPMI medium supplemented with 5% heat-inactivated fetal bovine serum (unstimulated) or the same growth medium supplemented with the phorbol ester PMA (Sigma-Aldrich, St. Louis, MO) at a concentration of 20 ng/mL (stimulated). Stable identify of this clonal line is continuously monitored by assessing hallmark changes in splicing induced by PMA (*Cole et al., 2015*; *Martinez et al., 2012*; *Shankarling et al., 2013*).

Low cycle reverse transcription-PCR (RT-PCR) was performed on 0.5 micrograms of RNA as described previously in detail (*Rothrock et al., 2003*) using sequence specific primers. Gels were quantified by densitometry with the use of a Typhoon PhosphorImager (Amersham Biosciences, UK). Primers and expected size of products for all events are given in *Supplementary file 2*.

For cassette exon LSVs percent spliced in was calculated as the percent of isoforms including the alternative exon over the total inclusion and exclusion isoforms. For complex LSVs, each band present on the gel was quantified. The percent selected index (PSI) for each junction of an LSV was calculated as the isoform(s) including that junction over the total isoforms present. For example, for the *Camk2g* source LSV (*Figure 1C*) the percent usage of the red junction that goes from the reference source exon 14 to exon 15 corresponds to the sum of the bands corresponding to the 214 nt isoform that includes exon 15 alone and the 256 nt isoform that includes both exons 15 and 16.

## Software availability

MAJIQ and VOILA are available for download at majiq.biociphers.org

## Acknowledgements

The authors would like to thank Dan Rader, Steve Liebhaber, and Chris Stoeckert for helpful feedback on preliminary versions of the manuscript. This research was supported by NIH grants R01 AG046544 to YB, R01 GM067719 to KWL, and a pilot grant to YB from Penn Medicine Neuroscience Center. The Accelerating Medicines Partnership for Alzheimer's Disease (AMP-AD) Target Discovery Consortium data were generated from postmortem brain tissue collected through the Mount Sinai VA Medical Center Brain Bank and were provided by Dr. Eric Schadt from Mount Sinai School of Medicine. This data can be accessed at doi:10.7303/syn2580853.

## Additional information

### Funding

| Funder | Grant reference number | Author |
| --- | --- | --- |
| National Institute on Aging | AG046544 | Yoseph Barash |
| National Institute of General Medical Sciences | GM067719 | Kristen W Lynch |
| The Penn Medicine Neuroscience Center | | Yoseph Barash |

The funders had no role in study design, data collection and interpretation, or the decision to submit the work for publication.

### Author contributions

JVG, Developed MAJIQ, Performed the species analysis, Performed PSI and dPSI quantification accuracy analysis, Performed LSV genome wide analysis and conservation analysis, Developed an early version of MAJIQ, Input to develop the algorithm, Acquisition of data, Analysis and

interpretation of data, Drafting or revising the article; AB, Developed VOILA, Performed the species analysis, Performed PSI and dPSI quantification accuracy analysis, Performed the protein feature enrichment analysis, Acquisition of data, Analysis and interpretation of data, Drafting or revising the article; MRG, Performed case study analyses and RT-PCR assays, Performed PSI and dPSI quantification accuracy analysis, Input to the protein feature enrichment analysis, Performed the analysis of AD patients and the multi-dataset meta analysis, Acquisition of data, Analysis and interpretation of data, Drafting or revising the article; JGonzal-V, Developed an early version of MAJIQ, Acquisition of data, Analysis and interpretation of data; NFL, Extracted RNA, Acquisition of data, Contributed unpublished essential data or reagents; JBH, Supervised the extraction of RNA, Acquisition of data, Contributed unpublished essential data or reagents; KWL, Supervised experiments and analyses, Acquisition of data, Drafting or revising the article, Contributed unpublished essential data or reagents; YB, Developed the algorithm, Conception and design, Analysis and interpretation of data, Drafting or revising the article

**Author ORCIDs**
Yoseph Barash, http://orcid.org/0000-0003-3005-5048

## Additional files

### Supplementary files
• Supplementary file 1. Protein features enrichment analysis results.
• Supplementary file 2. List of primers used for experimental validation.

### Major datasets
The following previously published datasets were used:

| Author(s) | Year | Dataset title | Dataset URL | Database, license, and accessibility information |
|---|---|---|---|---|
| Keane TM, Goodstadt L, Danecek P, White MA, Wong K, Yalcin B, Heger A, Agam A, Slater G, Goodson M, Furlotte NA, Eskin E, Nellåker C, Whitley H, Cleak J, Janowitz D, Hernandez-Pliego P, Edwards A, Belgard TG, Oliver PL, McIntyre RE, Bhomra A, Nicod J, Gan X, Yuan W, van der Weyden L, Steward CA, Bala S, Stalker J, Mott R, Durbin R, Jackson IJ, Czechanski A, Guerra-Assunção JA, Donahue LR, Reinholdt LG, Payseur BA, Ponting CP, Birney E, Flint J, Adams DJ | 2011 | Mouse Genome Project | http://www.ebi.ac.uk/ena/data/view/ERP000591&display=html | Publicly available at the EBI European Nucleotide Archive (Accession no: ERP000591). |
| Zhang R, Lahens NF, Ballance HI | 2014 | A circadian gene expression atlas in mammals: Implications for biology and medicine | http://www.ncbi.nlm.nih.gov/geo/query/acc.cgi?acc=GSE54652 | Publicly available at the NCBI Gene Expression Omnibus (Accession no: GSE54652). |

| | | | | |
|---|---|---|---|---|
| Barbosa-Morais NL, Irimia M, Pan Q, Xiong HY, Gueroussov S, Lee LJ, Slobodeniuc V, Kutter C, Watt S, Colak R, Kim T, Misquitta-Ali CM, Wilson MD, Kim PM, Odom DT, Frey BJ, Blencowe BJ | 2012 | The Evolutionary Landscape of Alternative Splicing in Vertebrate Species | http://www.ncbi.nlm.nih.gov/geo/query/acc.cgi?acc=GSE41338 | Publicly available at the NCBI Gene Expression Omnibus (Accession no: GSE41338). |
| Brawand D, Soumillon M, Necsulea A, Julien P, Csárdi G, Harrigan P, Weier M, Liechti A, Aximu-Petri A, Kircher M, Albert FW, Zeller U, Khaitovich P, Grützner F, Bergmann S, Nielsen R, Pääbo S, Kaessmann H | 2011 | The evolution of gene expression levels in mammalian organs | http://www.ncbi.nlm.nih.gov/geo/query/acc.cgi?acc=GSE30352 | Publicly available at the NCBI Gene Expression Omnibus (Accession no: GSE30352). |
| Schroth GP | 2011 | Illumina Human Body Map 2.0 Project | http://www.ncbi.nlm.nih.gov/geo/query/acc.cgi?acc=GSE30611 | Publicly available at the NCBI Gene Expression Omnibus (Accession no: GSE30611). |
| Bai B, Hales CM, Chen PC, Gozal Y, Dammer EB, Fritz JJ, Wang X, Xia Q, Duong DM, Street C, Cantero G, Cheng D, Jones DR, Wu Z, Li Y, Diner I, Heilman CJ, Rees HD, Wu H, Lin L, Szulwach KE, Gearing M, Mufson EJ, Bennett DA, Montine TJ, Seyfried NT, Wingo TS, Sun YE, Jin P, Hanfelt J, Willcock DM, Levey A, Lah JJ, Peng J | 2013 | U1 small nuclear ribonucleoprotein complex and RNA splicing alterations in Alzheimer's disease | http://www.ncbi.nlm.nih.gov/sra/?term=SRP056863 | Publicly available at the NCBI Short Read Archive (Accession no: SRP056863). |
| Yan Q, Weyn-Vanhentenryck SM, Wu J, Sloan SA, Zhang Y, Chen K, Wu JQ, Barres BA, Zhang C | 2015 | Systematic discovery of regulated and conserved alternative exons in the mammalian brain reveals NMD modulating chromatin regulators | http://www.ncbi.nlm.nih.gov/sra/SRP055008 | Publicly available at the NCBI Short Read Archive (Accession no: SRP055008). |
| Giudice J, Xia Z, Wang ET, Scavuzzo MA, Ward AJ, Kalsotra A, Wang W, Wehrens XH, Burge CB, Li W, Cooper TA | 2014 | Transcriptome modulation of ventricles, cardiomyocytes and cardiac fibroblasts during postnatal mouse development | http://www.ncbi.nlm.nih.gov/geo/query/acc.cgi?acc=GSE49906 | Publicly available at the NCBI Gene Expression Omnibus (Accession no: GSE49906). |
| Bhate A, Parker DJ, Bebee TW, Ahn J, Arif W, Rashan EH, Chorghade S, Chau A, Lee JH, Anakk S, Carstens RP, Xiao X, Kalsotra A | 2015 | ESRP2 Regulates A Conserved And Cell-Type-Specific Splicing Program to Support Postnatal Liver Maturation | http://www.ncbi.nlm.nih.gov/geo/query/acc.cgi?acc=GSE67009 | Publicly available at the NCBI Gene Expression Omnibus (Accession no: GSE67009). |

| | | | | |
|---|---|---|---|---|
| Singh RK, Xia Z, Bland CS, Kalsotra A, Ruddy M, Curk T, Ule J, Li W, Cooper TA | 2014 | Rbfox2-coordinated alternative splicing of Mef2d and Rock2 controls myoblast fusion during myogenesis | http://www.ncbi.nlm.nih.gov/geo/query/acc.cgi?acc=GSE58928 | Publicly available at the NCBI Gene Expression Omnibus (Accession no: GSE58928). |
| Quesnel-Vallières M, Irimia M, Cordes SP, Blencowe BJ | 2015 | Essential roles for the splicing regulator nSR100/SRRM4 during nervous system development | http://www.ncbi.nlm.nih.gov/geo/query/acc.cgi?acc=GSE65818 | Publicly available at the NCBI Gene Expression Omnibus (Accession no: GSE65818). |
| Li Q, Zheng S, Han A, Lin CH, Stoilov P, Fu XD, Black DL | 2013 | Alternative splicing in PTBP2 knockout mouse brain | http://www.ncbi.nlm.nih.gov/geo/query/acc.cgi?acc=GSE51740 | Publicly available at the NCBI Gene Expression Omnibus (Accession no: GSE51740). |
| Lovci MT, Ghanem D, Marr H, Arnold J, Gee S, Parra M, Liang TY, Stark TJ, Gehman LT, Hoon S, Massirer KB, Pratt GA, Black DL, Gray JW, Conboy JG, Yeo GW | 2013 | Rbfox proteins regulate alternative mRNA splicing through evolutionarily conserved RNA bridges. | http://www.ncbi.nlm.nih.gov/sra/?term=SRP030031 | Publicly available at the NCBI Gene Expression Omnibus (Accession no: SRP030031). |
| Wang ET, Cody NA, Jog S, Biancolella M, Wang TT, Treacy DJ, Luo S, Schroth GP, Housman DE, Reddy S, Lécuyer E, Burge CB | 2012 | Transcriptome-wide Regulation of Splicing and mRNA Localization by Muscleblind Proteins | http://www.ncbi.nlm.nih.gov/geo/query/acc.cgi?acc=GSE39911 | Publicly available at the NCBI Gene Expression Omnibus (Accession no: GSE39911). |
| Charizanis K, Lee KY, Batra R, Goodwin M, Zhang C, Yuan Y, Shiue L, Cline M, Scotti MM, Xia G, Kumar A, Ashizawa T, Clark HB, Kimura T, Takahashi MP, Fujimura H, Jinnai K, Yoshikawa H, Gomes-Pereira M, Gourdon G, Sakai N, Nishino S, Foster TC, Ares M Jr, Darnell RB, Swanson MS | 2012 | Muscleblind-Like 2 mediated alternative splicing in the developing bain by mRNA sequencing | http://www.ncbi.nlm.nih.gov/geo/query/acc.cgi?acc=GSE38497 | Publicly available at the NCBI Gene Expression Omnibus (Accession no: GSE38497). |
| Yang J, Hung LH, Licht T, Kostin S, Looso M, Khrameeva E, Bindereif A, Schneider A, Braun T | 2014 | RBM24 is a major regulator of muscle-specific alternative splicing | http://www.ncbi.nlm.nih.gov/sra/?term=SRP044097 | Publicly available at the NCBI Gene Expression Omnibus (Accession no: SRP044097). |
| Wei N, Cheng Y, Wang Z, Liu Y, Luo C, Liu L, Chen L, Xie Z, Lu Y, Feng Y | 2015 | Next Generation Sequencing Analysis of Wild Type and SRSF10-/- embryonic day 13.5 heart Transcriptomes | http://www.ncbi.nlm.nih.gov/geo/query/acc.cgi?acc=GSE66965 | Publicly available at the NCBI Gene Expression Omnibus (Accession no: GSE66965). |
| Ye J, Beetz N, O'Keeffe S, Tapia JC, Macpherson L, Chen WV, Bassel-Duby R, Olson EN, Maniatis T | 2015 | hnRNP U protein is required for normal pre-mRNA splicing and postnatal heart development and function | http://www.ncbi.nlm.nih.gov/geo/query/acc.cgi?acc=GSE68178 | Publicly available at the NCBI Gene Expression Omnibus (Accession no: GSE68178). |

| Lee J, Damianov A, Lin CH, Fontes M, Parikshak NN, Anderson ES, Geschwind DH, Black DL, Martin KC | 2015 | Gene expression profiling of neurons with Rbfox1 and Rbfox3 knockdown and rescue with cytoplasmic or nuclear Rbfox1 isoform [RNA-seq] | http://www.ncbi.nlm.nih.gov/geo/query/acc.cgi?acc=GSE71916 | Publicly available at the NCBI Gene Expression Omnibus (Accession no: GSE71916). |
|---|---|---|---|---|
| Cole BS, Tapescu I, Allon SJ, Mallory MJ, Qiu J, Lake RJ, Fan HY, Fu XD, Lynch KW | 2015 | Global analysis of physical and functional RNA targets of hnRNP L reveals distinct sequence and epigenetic features of repressed and enhanced exons | http://www.ncbi.nlm.nih.gov/Traces/sra/sra.cgi?study=SRP059357 | Publicly available at the NCBI Gene Expression Omnibus (Accession no: SRP059357). |
| Wang ET, Ward AJ, Cherone JM, Giudice J, Wang TT, Treacy DJ, Lambert NJ, Freese P, Saxena T, Cooper TA, Burge CB | 2014 | Functional Antagonism Between CELF and Mbnl Proteins in Cytoplasm and Nucleus | http://www.ncbi.nlm.nih.gov/geo/query/acc.cgi?acc=GSE61893 | Publicly available at the NCBI Gene Expression Omnibus (Accession no: GSE61893). |
| Han A, Black DL | 2014 | Gene Expression and Exon Splicing Change Analysis of Mouse N2A Cell Transcriptome upon Polypyrimidine tract-binding protein depletion (II) | http://www.ncbi.nlm.nih.gov/geo/query/acc.cgi?acc=GSE52856 | Publicly available at the NCBI Gene Expression Omnibus (Accession no: GSE52856). |
| Bebee TW, Park JW, Sheridan KI, Warzecha CC, Cieply BW, Rohacek AM, Xing Y, Carstens RP | 2015 | Knockout mice reveal an essential role for Epithelial splicing regulatory proteins (Esrps) in mammalian development and epithelial splicing in vivo | http://www.ncbi.nlm.nih.gov/geo/query/acc.cgi?acc=GSE64357 | Publicly available at the NCBI Gene Expression Omnibus (Accession no: GSE64357). |
| Jangi M, Boutz PL, Paul P, Sharp PA | 2014 | Rbfox2 controls autoregulation in RNA binding protein networks | http://www.ncbi.nlm.nih.gov/geo/query/acc.cgi?acc=GSE54794 | Publicly available at the NCBI Gene Expression Omnibus (Accession no: GSE54794). |
| Schadt E, The Accelerating Medicines Partnership for Alzheimer's Disease (AMP-AD) Target Discovery Consortium | 2015 | Mount Sinai Brain Bank (MSBB) RNA sequencing study | https://www.synapse.org/#!Synapse:syn3157743 | Publicly available at the AMP AD Knowledge Portal (Accession no: syn3157743). |
| Zhang R, Lahens NF, Ballance HI, Hughes ME, Hogenesch JB | 2014 | A circadian gene expression atlas in mammals: implications for biology and medicine | http://www.ncbi.nlm.nih.gov/geo/query/acc.cgi?acc=GSE54651 | Publicly available at the NCBI Gene Expression Omnibus (Accession no: GSE54651). |

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
