## [Decision Letter]

Thank you for submitting your work entitled "A new view of transcriptome complexity
and regulation through the lens of local splicing variations" for consideration by
*eLife*. Your article has been reviewed by three peer reviewers, one
of whom is a member of our Board of Reviewing Editors, and the evaluation has been
overseen by Juan Valcárcel (Reviewing Editor) and James Manley as the Senior Editor. One
of the other two reviewers, Roderic Guigo, has agreed to share his identity.

The reviewers have discussed the reviews with one another and the Reviewing editor has
drafted this decision to help you prepare a revised submission.

Summary:

The work described in the manuscript by Vaquero-Garcia et al. has four main
components:

1) A method to identify and categorize Alternative Splicing (AS) events-to which the
authors refer as to Local Splicing Variations (LSV).

2) A method (MAJIQ) to quantify these events using RNASeq data and a visualization
component (VOILA).

3) The application of the method to produce a catalogue of LSVs in mouse.

4) The detailed investigation of some of the complex LSVs identified that affect protein
domains in developmentally regulated genes and have roles in neurogenesis and other
brain functions. The authors illustrate the usefulness of their technique by reporting
novel tissue-specific transcript variants of Camk2d and a poison exon in Ptbp1, as well
as the apparent enrichment of certain transcript variants in brain samples from
Alzheimer disease patients.

The major advances of these tools are 1) that they are not restricted to being able to
analyze the standard types of splicing events (skipped exons, retained introns,
alternative 5' splice sites, etc.) but can also identify and quantify much more complex
local splicing variations, 2) can analyze both annotated and novel splicing events
identified using RNA-seq data, and 3) can compare samples using replicates. There is
indeed a need to understand the full complexity of transcript variants in most
biological/pathological settings, and therefore sensitive methods for their detection
and quantification, particularly using standard sequencing depth datasets, can be very
valuable. The performance of the package, which combines various softwares into a single
pipeline, appears to be excellent and all tools are available.

Essential revisions:

1) The manuscript would benefit from presenting the work in framework of previous
developments in the field, comparing with them, and emphasizing the relative merits of
the current contribution. The authors seem to imply that theirs is the first method that
attempts to systematically classify and categorize AS events, beyond merely, exon
skipping, alternative usage of splicing sites or intron retention. But, in my opinion,
this is incorrect. The authors appear to ignore a large body of previous literature that
actually attempts to address exactly this problem. Worth mentioning in this regard is
the work of Nagasaki et al. (2005, http://bioinformatics.oxfordjournals.org/content/22/10/1211.full) or
Sammeth et al. (2008, http://journals.plos.org/ploscompbiol/article?id=10.1371/journal.pcbi.1000147).
In the latter work, in particular, analysis similar to those presented in Figure 3 in Vaquero-Garcia et al., regarding the
frequency of complex LSVs depending on the annotation used and the species considered,
are presented. Moreover the methods developed by Sammeth et al. have actually been used
to categorize AS events in a number of publications (https://scholar.google.es/scholar?cites=8586199022962348855&as_sdt=2005&sciodt=0,5&hl=en).
Also single source and single target LSVs are quantified in their recent work by
Pervouchine et al. (2013, http://bioinformatics.oxfordjournals.org/content/29/2/273.long). The
authors also claim that they obtain splicing information by combining the annotation and
RNA-seq evidence (Figure 2). It is not clear if
they use only splice junctions, or also exons and transcript models. The latter is
important for the definition of LSV. Indeed, if "the reference exon may include
multiple 3' splice sites (ST-LSV) or 5' splice sites (SS-LSV)", as it is said in
Materials and methods, it would be necessary to know novel exons as continuous units. If
this is the case, a lot more has to be explained in Materials and methods regarding the
procedure of exon discovery from RNA-seq. In summary, in contrast to the claims by
Vaquero-Garcia et al., complex LSVs have received wide attention within the field.

2) Another potential caveat is whether the results reported will persuade the general
reader of the genuine potential of the approach to get biological insight.

Specifically:

A) The authors argue that complex LSVs represent over 30% of all transcripts variants.
The general reader may wonder whether the switches in transcript isoforms within this
category generally represent significantly large differences between tissues or
biological conditions as to have biological impact. In other words, is
biologically-relevant LSV likely to be limited to exceptional cases, or are the majority
of complex variants likely to be functionally relevant? It would be useful, in this
regard, to analyze the range and frequency distribution of ΔPSI values associated with
complex LSVs compared to more standard alternative splicing events. Analysis of the
conservation level of the predictably more relevant events could also be helpful.

B) The authors predict that their method "will advance the ability to relate
tissue-specific splice variation to genetic variation, phenotype and disease". They
illustrate this by the identification of transcript variants possibly related to
Alzheimer Disease, but the results are essentially descriptive and, considering the
number of samples tested and the absence of validation in different cohorts, of limited
value to extract rigorous biomedical conclusions. The manuscript would greatly benefit
from additional efforts to more broadly illustrate the utility of the tool, for example
to show how the approach is advantageous for explaining the impact of genetic variation
using publicly available data from the GTEx project (e.g. pilot samples), ENCODE,
modENCODE, etc.

[Editors' note: further revisions were requested prior to acceptance, as described
below.]

Thank you for resubmitting your work entitled "A new view of transcriptome
complexity and regulation through the lens of local splicing variations" for
further consideration at *eLife*. Your revised article has been favorably
evaluated by James Manley (Senior editor) and a Reviewing editor. The manuscript has
been improved but there are some remaining issues that need to be addressed before
acceptance, as outlined below:

The authors have significantly expanded and improved their manuscript, which can now
provide a generally useful tool for exploring transcriptome diversity in RNA-seq data
and, on these basis, would be acceptable for publication in *eLife*.

The authors have done a good job discussing previous efforts by other scientists to
identify complex alternative splicing decisions (main point #1 of the referees' report)
and to illustrate the range and frequency distribution of complex LSVs compared to more
standard alternative splicing events (point #2A). It would be helpful if the authors
could more clearly state what was the validation rate of their disease-associated LSV
changes in a second cohort of Alzheimer Disease patients (point #2B) (unless I have
missed it, they only provide the number of LSVs reproducibly differentially spliced –
p20 – but not what is the fraction of predictions validated in the two datasets).
Another point: it is not clearly stated in the text/Figure 5 legend that the data corresponds to LSVs consistently
differentially spliced between brain samples from patients and controls in the two
datasets (as stated in the rebuttal).

---

## [Author Response]

*1) The manuscript would* benefit *from presenting the work in
framework of previous developments in the field, comparing with them, and emphasizing
the relative merits of the current contribution. The authors seem to imply that
theirs is the first method that attempts to systematically classify and categorize AS
events, beyond merely, exon skipping, alternative usage of splicing sites or intron
retention. But, in my opinion, this is incorrect. The authors appear to ignore a
large body of previous literature that actually attempts to address exactly this
problem. Worth mentioning in this regard is the work of Nagasaki et al.
(2005,*

*http://bioinformatics.oxfordjournals.org/content/22/10/1211.full*)
*or Sammeth et al. (2008, http://journals.plos.org/ploscompbiol/article?id=10.1371/journal.pcbi.1000147).
In the latter work, in particular, analysis similar to those presented in Figure 3 in Vaquero-Garcia et al., regarding the
frequency of complex LSVs depending on the annotation used and the species
considered, are presented. Moreover the methods developed by Sammeth et al. have
actually been used to categorize AS events in a number of publications (https://scholar.google.es/scholar?cites=8586199022962348855&as_sdt=2005&sciodt=0,5&hl=en).
Also single source and single target LSVs are quantified in their recent work by
Pervouchine et al. (2013, http://bioinformatics.oxfordjournals.org/content/29/2/273.long). The
authors also claim that they obtain splicing information by combining the annotation
and RNA-seq evidence (Figure 2). It is not
clear if they use only splice junctions, or also exons and transcript models. The
latter is important for the definition of LSV. Indeed, if "the reference exon
may include multiple 3' splice sites (ST-LSV) or 5' splice sites (SS-LSV)", as
it is said in Materials and methods, it would be necessary to know novel exons as
continuous units. If this is the case, a lot more has to be explained in Materials
and methods regarding the procedure of exon discovery from RNA-seq. In summary, in
contrast to the claims by Vaquero-Garcia et al., complex LSVs have received wide
attention within the field.*

The original manuscript lacked a section describing related work, which was mistakenly
removed before submission. We apologize for this mistake and have now re introduced an
extended section for this in the revised Discussion. We also added text in the Methods
section to answer the question regarding how MAJIQ combines known annotation and de-novo
junction to annotate putative de-novo exonic regions.

*2) Another potential caveat is whether the results reported will persuade the
general reader of the genuine potential of the approach to get biological
insight.*

*Specifically:*

*A) The authors argue that complex LSVs represent over 30% of all transcripts
variants. The general reader may wonder whether the switches in transcript isoforms
within this category generally represent significantly large differences between
tissues or biological conditions as to have biological impact. In other words, is
biologically-relevant LSV likely to be limited to exceptional cases, or are the
majority of complex variants likely to be functionally relevant? It would be useful,
in this regard, to analyze the range and frequency distribution of ΔPSI values
associated with complex LSVs compared to more standard alternative splicing events.
Analysis of the conservation level of the predictably more relevant events could also
be helpful.*

The reviewers raise here an excellent point, which is also related to the concern raised
below about disease relevance. To address the concern about significance of complex LSVs
we followed the reviewers’ suggestion and created a new figure (Figure 5). Figure 5 is a
meta analysis of over 30 datasets and 243 RNA-Seq experiments that shows the significant
enrichment of complex LSVs in datasets that cover developmental stages, key splice
factors, and human disease (AD) across diverse tissues and cell lines. Figure 5 shows the changes in LSV inclusion levels
(dPSI) across these conditions, just as suggested by the reviewers. Finally, Figure 5 shows the suggested conservation analysis
with a significant increase in intronic conservation for differentially spliced complex
LSVs compared to their binary LSVs counterparts.

*B) The authors predict that their method "will advance the ability to
relate tissue-specific splice variation to genetic variation, phenotype and
disease". They illustrate this by the identification of transcript variants
possibly related to Alzheimer Disease, but the results are essentially descriptive
and, considering the number of samples tested and the absence of validation in
different cohorts, of limited value to extract rigorous biomedical conclusions. The
manuscript would greatly benefit from additional efforts to more broadly illustrate
the utility of the tool, for example to show how the approach is advantageous for
explaining the impact of genetic variation using publicly available data from the
GTEx project (e.g. pilot samples), ENCODE, modENCODE, etc.*

To address the concern about the anecdotal nature of the disease related results in the
original submission we performed the following: First, we followed the reviewers’
suggestion and validated these results on an independent data set (syn3157743 from the
AMP-AD Target Discovery Consortium data portal) which is much larger: 157 samples from
AS patient’s brains and 128 control samples, across three different brain sub regions.
Second, we added a genome wide analysis of LSVs that are consistently differentially
spliced between brain samples from patients and controls in the two datasets (included
in Figure 5 described above). We very much agree
this work is still just the tip of the iceberg and we believe much more can be done with
the AD datasets and many others. Specifically, the suggested analysis of genetic
variations with MAJIQ is a great direction to pursue in future work. However, we are
very much aware that a careful analysis of this sort is beyond the scope of this
paper.

[Editors' note: further revisions were requested prior to acceptance, as described
below.]

*[…] The authors have done a good job discussing previous efforts by other
scientists to identify complex alternative splicing decisions (main point #1 of the
referees' report) and to illustrate the range and frequency distribution of complex
LSVs compared to more standard alternative splicing events (point #2A). It would be
helpful if the authors could more clearly state what was the validation rate of their
disease-associated LSV changes in a second cohort of Alzheimer Disease patients
(point #2B) (unless I have missed it, they only provide the number of LSVs
reproducibly differentially spliced* – *p20* – *but not
what is the fraction of predictions validated in the two datasets). Another point: it
is not clearly stated in the text/Figure 5
legend that the data corresponds to LSVs consistently differentially spliced between
brain samples from patients and controls in the two datasets (as stated in the
rebuttal).*

We have worked hard to address the reviewers’ concerns and suggestions. In particular,
to address the reviewers’ concern about the significance of complex LSVs we performed a
whole new meta analysis of LSVs across 32 datasets covering diverse tissues, cell lines,
developmental stages and splice factors knockdown. We added this to a conservation
analysis as suggested by the reviewers, creating a new figure (Figure 5). In addition, to address the concern regarding the
relevance of our findings to disease studies we added a genome wide analysis of
differentially spliced LSVs in independent datasets of 285 samples from AD patients and
controls, which validated our initial findings for the CAMK2 family and other genes. We
also added a section detailing previous related work, which was erroneously omitted in
the original submission, a mistake we apologize for. Finally, we made sure to add
details in all sections indicated by the reviewers as unclear.